# Drug transport kinetics of intravascular triggered drug delivery systems

Timo L. M. ten Hagen[1], Matthew R. Dreher[2], Sara Zalba [1], Ann L. B. Seynhaeve[1], Mohamadreza Amin[1], Li Li[1] & Dieter Haemmerich [3,4✉]

Intravascular triggered drug delivery systems (IV-DDS) for local drug delivery include various stimuli-responsive nanoparticles that release the associated agent in response to internal (e.g., pH, enzymes) or external stimuli (e.g., temperature, light, ultrasound, electromagnetic fields, X-rays). We developed a computational model to simulate IV-DDS drug delivery, for which we quantified all model parameters in vivo in rodent tumors. The model was validated via quantitative intravital microscopy studies with unencapsulated fluorescent dye, and with two formulations of temperature-sensitive liposomes (slow, and fast release) encapsulating a fluorescent dye as example IV-DDS. Tumor intra- and extravascular dye concentration dynamics were extracted from the intravital microscopy data by quantitative image processing, and were compared to computer model results. Via this computer model we explain IV-DDS delivery kinetics and identify parameters of IV-DDS, of drug, and of target tissue for optimal delivery. Two parameter ratios were identified that exclusively dictate how much drug can be delivered with IV-DDS, indicating the importance of IV-DDS with fast drug release (~sec) and choice of a drug with rapid tissue uptake (i.e., high first-pass extraction fraction). The computational model thus enables engineering of improved future IV-DDS based on tissue parameters that can be quantified by imaging.

[1] Laboratory Experimental Oncology and Nanomedicine Innovation Center Erasmus (NICE), Department of Pathology, Erasmus Medical Center, Rotterdam, The Netherlands. [2] Boston Scientific, Marlborough, MA, USA. [3] Department of Pediatrics, Medical University of South Carolina, Charleston, SC, USA. [4] Department of Bioengineering, Clemson University, Clemson, SC, USA. ✉email: haemmerich@ieee.org

Drug delivery systems (DDS) have an array of potential clinical applications but are particularly relevant for cancer therapy. Traditional systemic chemotherapy is a mainstay of cancer treatment. Efficacy of chemotherapy is however limited by insufficient delivery of drugs to cancer cells[1] and by normal tissue toxicity that limits the dose that may be administered. To confront these challenges, various DDS such as liposomes[2], micelles[3], and macromolecular drug carriers[4,5] have been developed[6–11]. Most DDS are passive and accumulate in a tumor due to the unique tumor pathophysiology including greater vascular permeability and lack of functional lymphatics, termed the enhanced permeability and retention (EPR) effect[11–15]. Active or functionalized DDS, such as immunoliposomes, may have a specific affinity for target cells, which possibly improves retention and cellular uptake[16,17]. However, passive accumulation is still required before active targeting of cancer cells can occur[12]. While passive accumulation may be enhanced via targeting moieties and better retention, there is increasing consensus that alternate approaches are necessary as the EPR effect appears less effective in human tumors, and since EPR is highly heterogenous within and between tumors[18–20]. Alternative active means may be used to loco-regionally increase the bioavailability of a drug. A particular active strategy that is growing in popularity is stimuli-induced (i.e., triggered) intravascular release[12,21]. In this approach the DDS does not leave the vasculature, but the encapsulated drug is released inside the vessels and readily distributed into the targeted tissue (Fig. 1). The trigger that facilitates release can be inherent to the target site (e.g., pH, specific enzyme activity), or applied externally (e.g., temperature, light, ultrasound, electromagnetic fields, or X-rays)[22,23]. Candidates for the IV-DDS approach include various light-triggered nanoparticles[24–26], polymer-based DDS that release drugs triggered by hyperthermia[27] or pH[28,29], DDS that can be activated by

magnetic or electric fields[23,30], and liposomes triggered by X-ray radiation[31].

IV-DDS are perhaps best exemplified by temperature-sensitive liposomes (TSL). TSL are relatively stable at 37 °C, but actively release their drug payload when heated above the melting temperature of the lipids (~40–44 °C)[21,32–38]. When a solid tumor is subjected to localized hyperthermia, intravascular release of drug from TSL is triggered intratumorally resulting in localized delivery with up to 25 times higher tumor drug uptake than unencapsulated drug[21,33,38–40], while TSL circulating in non-heated tissues retain the drug. Another prevalent IV-DDS example are microbubbles, which release the associated drug when activated by ultrasound[41,42]. Various IV-DDS have been studied for several decades, but with only limited guidance on what the ideal properties of the delivery system or drug should be.

It is readily apparent that the stimuli-induced intravascular release approach has many variables that are spatiotemporally varying and interdependent[43] (Fig. S1). Importantly, the transport properties, such as plasma half-life and vascular permeability, depend on whether the drug is free (bioavailable), or associated with a DDS. Due to a large number of interdependent variables, computational modeling provides an effective means to explain and potentially optimize the stimuli-induced intravascular release paradigm used in IV-DDS.

Unfortunately, predictive capabilities of computer models are often limited by unknown parameter values requiring assumptions, and because of lack of adequate experimental validation of the computer models[44,45]. In this study, we identified the delivery kinetics of IV-DDS based on a computational model integrated with in vivo imaging studies where temperature-sensitive liposomes served as example IV-DDS. Via the computer model, we demonstrate interactions between DDS, drug, and physiology/biology. Importantly, all model parameters were experimentally

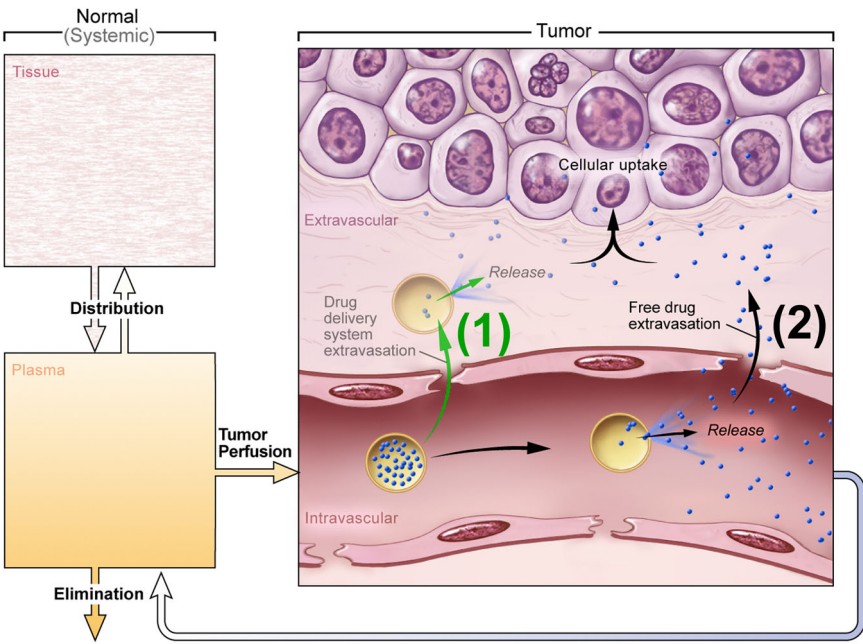

**Fig. 1 Schematic of intravascular triggered drug delivery systems (IV-DDS).** (1) Traditionally, DDS have been based on passive tumor targeting due to enhanced permeability and retention (EPR), where drug is released following extravasation of the DDS. (2) For IV-DDS, EPR is not relevant: IV-DDS enter the tumor microvasculature of the target region where the release trigger is present, and release the contained drug within the vasculature. The released drug extravasates rapidly into tissue and is then taken up by cancer cells. Both encapsulated and any released drug not taken up by the tumor are distributed systemically and eliminated, symbolized by the systemic plasma and tissue compartments at the left side.

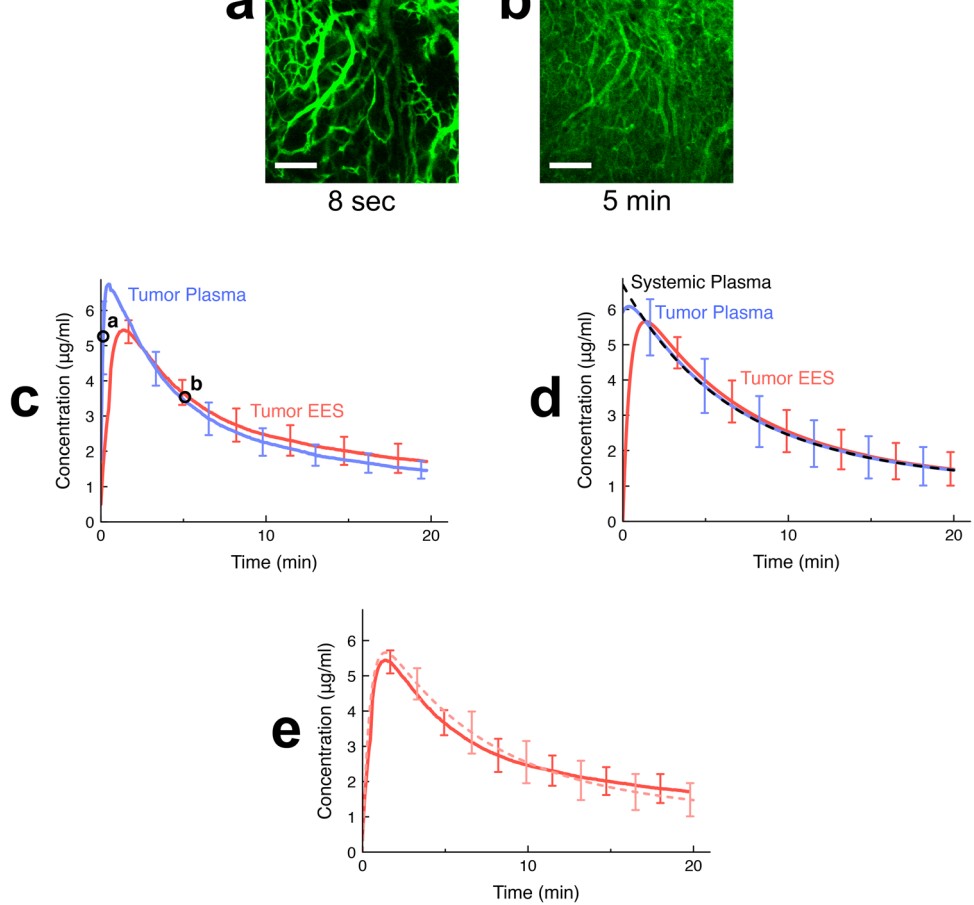

**Fig. 2 Delivery kinetics of unencapsulated dye in vivo and in computer model.** A bolus of unencapsulated fluorescent dye (carboxyfluorescein (CF)) was administered to tumor-bearing mice. **a** Intravital microscopy visualizes arrival of dye in the tumor vasculature, here shown 8 s after injection. Scale bar indicates 200 μm. **b** Uptake of dye by tumor extravascular-extracellular space (EES) is apparent, shown 5 min after injection. The whole image time series is available as Suppl. Movie 1. **c** Mean dye concentrations in plasma and EES in imaged tumor segment were extracted from imaging data (image acquisition every 4 s). Time points of microscopy images shown in (**a**, **b**) are indicated by circles. Error bars indicate standard deviation ($n = 3$ animals). **d** Computer simulations based on tumor transport parameters derived from in vivo studies accurately reproduce the delivery kinetics of unencapsulated (free) dye. Error bars indicate model uncertainty due to uncertainty of model parameters. The mean normalized error of tumor EES concentration of the computer model was 3.8%. **e** Direct comparison of tumor EES concentration between computer model (solid curve) and intravital data (faint dotted curve).

measured in vivo, and intravital fluorescence microscopy studies were used to validate the computational model.

## Results

**In vivo derived computer model accurately reproduces delivery kinetics of unencapsulated drug.** Computer models typically require a large number of parameter values[44]. These parameters are usually either estimated or extracted from prior publications[45] rather than directly measured in the experimental model used for validation. Here, we present a computer model based exclusively on in vivo measured parameters; furthermore, validation was performed in the same in vivo model. The computer model included all drug transport processes indicated in Fig. 1.

Intravital microscopy is commonly used to document the distribution of drugs in solid tumors, and is ideal for computer model validation as well as parameter extraction[46–48]. Recent imaging approaches additionally enable the visualization and monitoring of both nanoparticles and drug release at the cellular scale[24]. We used the fluorescent dye carboxyfluorescein (CF) as model drug, which is not taken up by cells. This was necessary (1) for the study of transvascular transport independent of cell

uptake, and (2) for the required direct conversion of fluorescence to concentration (Fig. S2), since any fluorescence changes due to drug interactions with intracellular components are averted. We performed studies in mice where we imaged a tumor segment during and after administration of unencapsulated dye (Fig. 2a, b, Suppl. Movie 1). We used image-processing techniques to extract the time course of intra- and extravascular dye fluorescence (Fig. S3), and converted fluorescence to absolute dye concentration (Fig. 2c) by calibration with quantified plasma samples (Fig. S2). From these calibrated data (Fig. 2c), we determined the tumor transport parameters and from additional studies in mice with sequential blood sampling following dye injection, we determined the systemic distribution and elimination parameters by pharmacokinetic modeling (Fig. S4–S7). This provided us with a complete parameter set for the computer model (Table 1). A direct comparison between computer model results (Fig. 2d) and experiments (Fig. 2c) demonstrates the ability to simulate delivery kinetics of unencapsulated drug at very high accuracy (see Fig. 2e, Table S1), likely in part because the majority of the transport parameters (Table 1) were measured in the same animals and tumor segments in which validation studies were performed (Fig. 6).

**Table 1 In vivo measured tumor transport parameters and pharmacokinetic parameters.**

| Parameter | Value | Parameter description | Source |
|---|---|---|---|
| *Tumor transport parameters* | | | |
| $v_p$ | 0.23 ± 0.07 | Tumor plasma volume fraction (plasma represents 23% of tumor tissue) | Intravital Imaging |
| $v_{e,av}$ | 0.28 ± 0.13 | Available extravascular tumor volume fraction (28% of extravascular volume is available to dye for distribution) | Intravital Imaging |
| $TT$ | 5.0 ± 0.5 s | Tissue transit time of tumor segment (time that plasma requires to pass through the imaged tumor segment) | Intravital Imaging |
| $PS^{\#}$ | 0.012 ± 0.005 s$^{-1}$ | Vascular permeability-surface area product (determines how fast dye transport between plasma and extravascular space occurs) | Intravital Imaging |
| $F_p$ | 0.047 ± 0.02 mL/(mL s) | Tumor plasma perfusion rate | Calculated from $v_p$, $TT$ |
| $EF^{\#}$ | 0.22 | First-pass extraction fraction (22% of dye is extracted by tissue during first pass) | Calculated from $PS$, $F_p$ |
| *Pharmacokinetic parameters* | | | |
| $V_D^{\#}$ | 8.99 ± 0.58 mL | Initial volume of distribution (Peak plasma concentration = (Injected dose)/$V_D$) | Serial blood sampling |
| $k_p^{\#}$ | 1.29 ± 0.52 × 10$^{-3}$ s$^{-1}$ | Transport rate constant, tissue → plasma | Serial blood sampling |
| $k_t^{\#}$ | 0.90 ± 0.22 × 10$^{-3}$ s$^{-1}$ | Transport rate constant, plasma → tissue | Serial blood sampling |
| $k_e^{\#}$ | 0.80 ± 0.30 × 10$^{-3}$ s$^{-1}$ | Elimination rate constant | Serial blood sampling |

Tumor transport parameters were measured from intravital imaging studies, and pharmacokinetic parameters are based on additional serial blood sampling studies (Fig. 6). The tumor transport parameters are specific to the tumor segment visible under imaging and may not be representative of the whole tumor.
#Parameters are specific to the employed drug/dye, here carboxyfluorescein (CF).

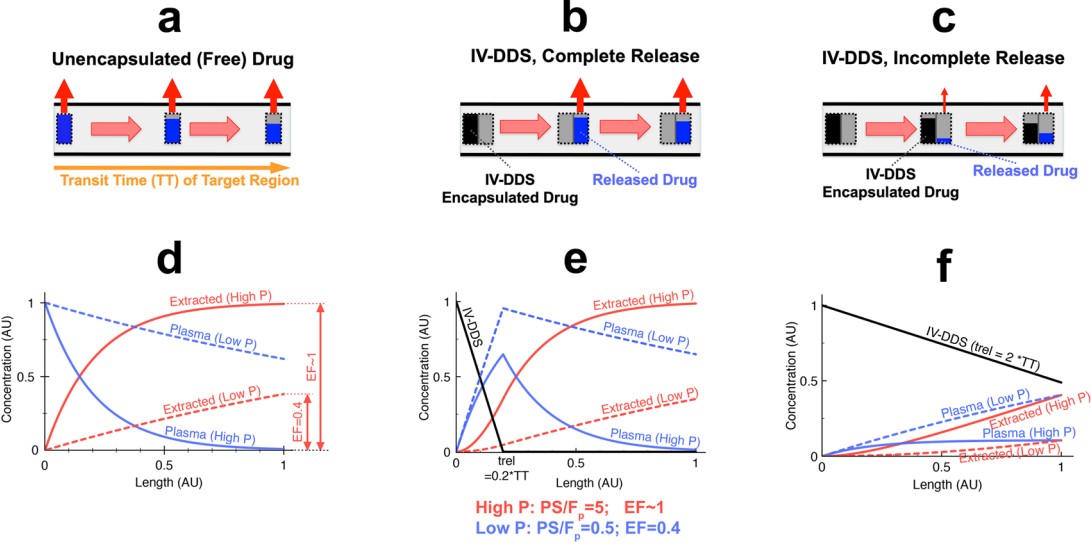

**Fig. 3 Schematic of microvascular concentration gradient.** Plasma traverses the microvasculature between supplying artery and draining vein of the target tissue segment. Plasma concentration of unencapsulated/released drug (blue bar), and IV-DDS-encapsulated drug (black bar) are shown, with red arrows indicating tissue drug uptake (transvascular transport into interstitium/EES). Three cases are presented: **a** Unencapsulated drug infusion, **b** IV-DDS with complete release during transit ($t_{rel} \ll TT$), and **c** IV-DDS with incomplete release ($t_{rel} \gg TT$). **d–f** For each case (**a–c**), the corresponding longitudinal concentration gradients along microvasculature length are shown based on computer models: free drug plasma concentration (blue), cumulative extracted amount by tissue (red), and IV-DDS-encapsulated drug plasma concentration (black). Both a drug with high permeability that is rapidly extracted ("High P", solid lines; $PS/F_p = 5$; $EF$~1), and a drug with lower permeability and slow extraction ("Low P", dashed lines; $PS/F_p = 0.5$; $EF = 0.4$) are presented. Note that all figures show the first pass where no drug has yet been extracted (i.e. interstitial/EES concentration is zero).

**Microvascular drug concentration gradient explains delivery kinetics of IV-DDS.** In traditional chemotherapy, drug is administered as a bioavailable ("free") agent, and the drug is extracted by tissue (transvascular transport into extravascular space) as the drug passes through the microvasculature of both targeted and untargeted tissue regions (Fig. 3a). The amount of drug extracted by tissue depends on vascular drug permeability and perfusion, where in the case of high permeability (solid curves in Fig. 3d), complete extraction of the drug may occur. The extent of drug extraction is often quantified by the parameter "first-pass extraction fraction (or extraction ratio)" (*EF*), which in

turn depends on plasma perfusion ($F_p$[1/s]), vascular drug permeability ($P$[cm/s]), and vascular surface area ($S$[cm$^2$/cm$^3$])[49]:

$$EF = 1 - e^{-PS/F_p} \qquad (1)$$

The latter two parameters are often combined as "vascular permeability-surface area product" ($PS$[1/s]), and we experimentally measured (via intravital microscopy) and used this parameter product in our computer model. The case of complete extraction ($EF \sim 1$, equivalent to $F_p \ll PS$) is commonly termed "perfusion-limited transport" since tissue drug uptake is limited

by perfusion ($F_p$). The case of incomplete extraction ($EF \ll 1$, equivalent to $PS \ll F_p$) is usually termed "permeability-limited transport" since tissue drug uptake is limited by vascular permeability ($P$)[49,50]. Another important parameter is the "tissue transit time" ($TT$[s]), and describes the time required for plasma to pass through the microvasculature of the tissue segment of interest. The transit time ($TT$) is related to plasma perfusion ($F_p$) via plasma volume fraction ($v_p$):

$$TT = \frac{v_p}{F_p} \qquad (2)$$

For IV-DDS the delivery kinetics are different than for free (unencapsulated) drug. Drug only becomes bioavailable after release from IV-DDS within the target regions microvasculature, initiated by the release trigger present in the target region (e.g., tumor)[20,21]. In the first scenario of IV-DDS with rapid release, complete drug release occurs shortly after IV-DDS enter the vessel (Fig. 3b, e). Note that the microvascular concentration gradient of free (released) drug (blue bars) in Fig. 3b is similar to Fig. 3a. That means that rapid-release IV-DDS are equivalent to infusion of bioavailable drug at high concentration for extended duration directly into the tumor vessels. In this paradigm, the body serves as a large reservoir of non-bioavailable drug that becomes 100% bioavailable once entering the targeted region. In the IV-DDS complete release scenario (Fig. 3b, e), the drug extraction approaches the free drug paradigm (Fig. 3a, d) yielding near maximal drug exposure to the cancer cells. In the incomplete release scenario where IV-DDS release the drug slowly (Fig. 3c, f), tissue drug uptake is greatly diminished due to the lower plasma concentration of bioavailable drug. Since the rate of drug extraction depends on the drug concentration difference between plasma and extravascular-extracellular space (EES), the higher plasma concentration of free (released) drug for rapidly releasing IV-DDS allow for greater drug extraction (Fig. 3b, e) compared to slowly releasing IV-DDS (Fig. 3c, f). Thus, the time required for drug release from IV-DDS is of primary importance; complete release during the transit of IV-DDS through the target regions microvasculature requires that the IV-DDS release time (=time to complete release, $t_{rel}$) is smaller than the tissue transit time ($TT$). In addition to IV-DDS release time, the extraction rate of the drug - once released (extraction fraction ($EF$)) - will impact delivery, with complete extraction ($EF \sim 1$) being desirable. Here, we assume zero-order IV-DDS release kinetics with a constant IV-DDS release rate (Fig. 3e, f; Fig. S8). However, the general discussion and results below are independent of the specific IV-DDS release kinetics (zero-order, first-order, etc.).

**Intravital microscopy studies and computer models demonstrate importance of rapid drug release**. We expanded the computer model simulating delivery of unencapsulated drug to predict IV-DDS delivery. This IV-DDS model was based on the parameters acquired in the prior in vivo studies with unencapsulated dye (Table 1, Table S2, Fig. 6, Fig. S9), as well as using in vitro measured IV-DDS release kinetics (Figs. S10–S12, Table S3). No additional in vivo parameters were measured from the in vivo experiments used for model validation, i.e. the model used to simulate IV-DDS was truly predictive (Fig. 7).

For in vivo studies, we employed temperature-sensitive liposomes (TSL) as representative IV-DDS example. To confirm the importance of rapid drug release suggested earlier (Fig. 3), we formulated two types of TSLs with either fast or slow release rate (fTSL or sTSL) (Figs. S10–S12), where fTSL released about 8 times as fast as sTSL (Table S3). After intravenous TSL administration, we exposed the tumor for ~5 min to hyperthermia (41–42 °C, Fig. 4a) under continuous imaging and converted

fluorescence to intra- and extravascular drug concentration based on image processing methods (Fig. S3). Once adequate temperature was achieved, intravascular release occurred, followed by tissue dye uptake (Fig. 4b, Suppl. Movies 2, 3). During the first few minutes, temperature increases towards 42 °C with concurrent decrease in TSL release time (Table S3), producing a progressive increase in plasma concentration of unencapsulated dye (Fig. 4c). EES concentration equilibrates with plasma concentration based on transvascular dye diffusion. For both fTSL and sTSL, a plateau concentration is reached (both in plasma and EES) towards the end of the heating period ($c_{plateau}$; see Fig. 4c, d). This plateau is limited by the amount of dye released during microvascular transit, and thus is higher for fTSL with shorter release time than sTSL. Once heating discontinues, TSL release stops, and back-diffusion of dye from EES into plasma occurs.

Based on the in vivo derived computer model for IV-DDS we performed simulations based on the parameters from in vivo studies with free dye (Table 1, Table S2), and considered further in vitro measured TSL release times at varying temperatures (Figs. S10–S12, Table S3) combined with in vivo temperature data (Fig. 4a). Figure 4d shows computer model results for both sTSL and fTSL, validating the ability to predict the overall in vivo delivery kinetics of both TSL formulations (Table S1). Compared to the intravital data, the computer model predicted a lower peak with wider plateau for fTSL. Further, for both sTSL and fTSL the computer model predicted a more rapid washout after heating stops that was less pronounced in vivo, especially for sTSL (Fig. 4e). Possible explanations for these observed differences include: (1) presence of an additional mechanism relevant at low concentrations (e.g., systemic leakage from TSL, cell uptake, or non-linear binding to plasma and/or tissue constituents); (2) TSL temperature dependence of release kinetics may differ in vivo from the in vitro data on which the model is based; (3) inaccuracies in tumor temperature measurements; (4) changes in tumor parameters in response to hyperthermia (e.g. perfusion).

The faster releasing fTSL resulted in ~10 times higher plateau concentration ($c_{plateau}$) compared to sTSL, both in experiments and computer model (Fig. 4c, d), while also showing that only ~40% of the maximum possible drug uptake was achieved. The maximum possible uptake requires complete release from TSL during tumor transit (Fig. 3b, e), followed by complete uptake. Thus, the maximum concentration ($c_{max}$) achievable in tumor plasma and tumor EES is equal to the plasma concentration of encapsulated drug in systemic circulation (i.e. before any release). It is apparent that the plateau concentration ($c_{plateau}$) is directly tied to release time ($t_{rel}$), or more specifically to the ratio between release time and tissue transit time ($t_{rel}/TT$, compare Fig. 3).

**Computer model predicts IV-DDS efficacy based on interplay of drug, physiology, and delivery system**. In addition to IV-DDS release time ($t_{rel}$), Fig. 3 suggests that the rate of drug extraction ($EF$, which depends on the ratio $PS/F_p$; see Eq. 1) impacts delivery. Careful analysis of the computer model equations reveals that the plateau concentration ($c_{plateau}$; see Fig. 4d) depends exclusively on two parameter ratios: $t_{rel}/TT$ and $PS/F_p$. Thus, we define two indices based on these ratios: (1) release index, $R.I. = t_{rel}/TT$; and (2) permeability index, $P.I. = PS/F_p$. These two indices include parameters depending on tumor physiology (vascular surface area ($S$), plasma perfusion ($F_p$), tissue transit time ($TT$)), the interaction between physiology and drug (vascular drug permeability ($P$)), and the interaction of drug with IV-DDS (release time ($t_{rel}$)) (compare Fig. S1). Notably, $PS$, $TT$, and $t_{rel}$ were also identified as dominant parameters in a global sensitivity analysis of all model parameters (Suppl. Note 1,

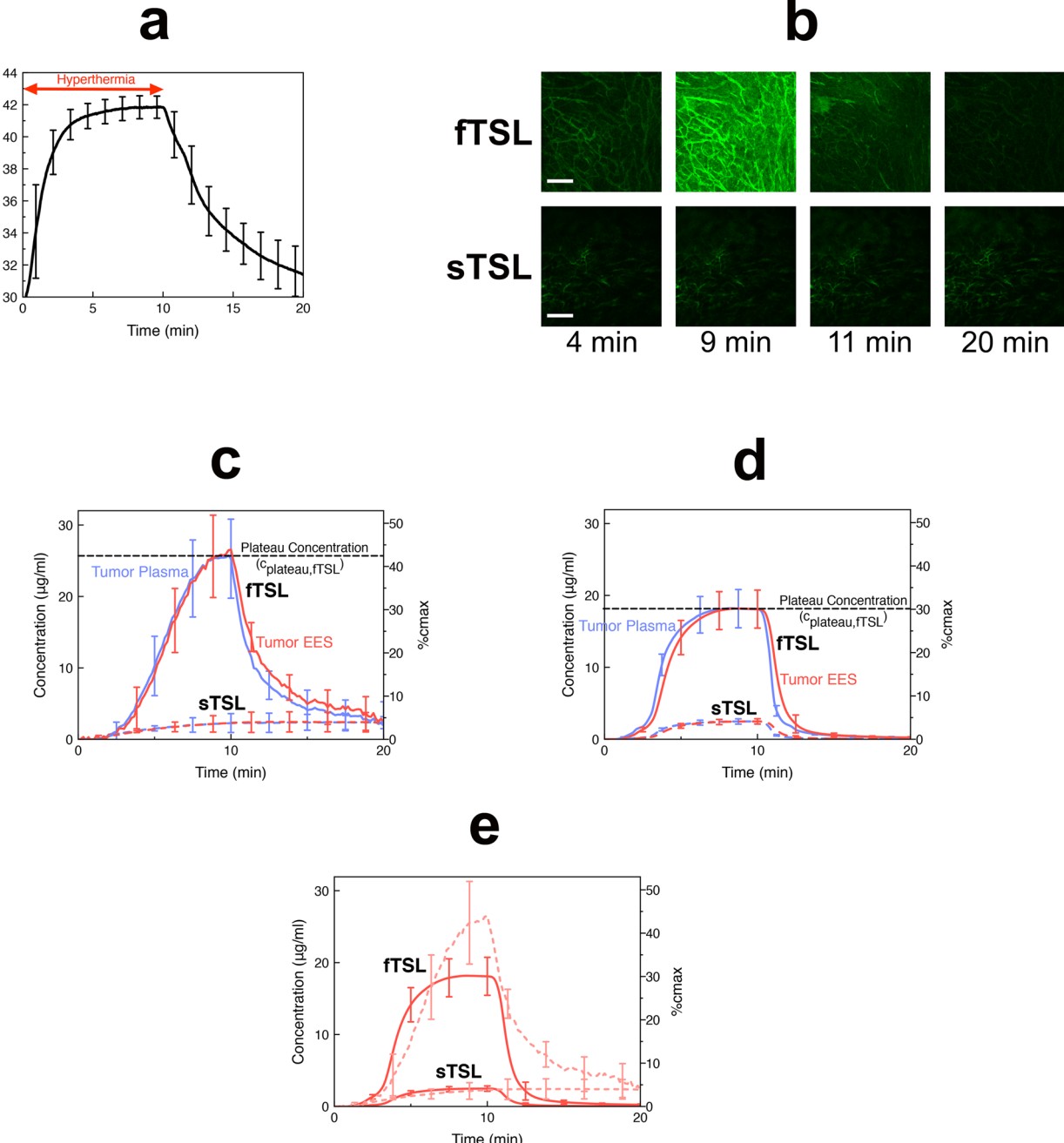

**Fig. 4 IV-DDS delivery kinetics in vivo and in computer model. a** Tumor temperature during hyperthermia exposure (41–42 °C) after administration of either sTSL or fTSL filled with fluorescent dye (carboxyfluorescein(CF)). Temperature was sampled every 2 s. **b** Intravital microscopy at 4 min visualizes intravascular release starting at ~40 °C, visible because of dequenching of dye fluorescence when released from TSL. Tissue uptake is highest just before heating stops (9 min), with rapid washout once heating has been discontinued (11 min). The whole image time series for sTSL and fTSL are available as Suppl. Movies 2, 3. Scale bar indicates 200 μm. **c** In vivo plasma and EES concentration time courses indicate that fTSL (solid curves) are about ten times more effective than sTSL (dashed curves). Graph shows mean concentrations in imaged tumor segment derived from intravital image data (image acquisition every 10 s). Error bars indicate standard deviation ($n = 3$ animals for both sTSL and fTSL). **d** Computer models based on in vivo tumor transport parameters and considering temperature-dependent TSL release time based on in vitro measurements. Error bars indicate model uncertainty due to uncertainty of model parameters. **e** Direct comparison of tumor EES concentration between computer model (solid curves) and intravital data (faint dotted curves). The peak concentration (plateau concentration ($c_{plateau}$)) in plasma and EES, indicated for fTSL by a black dashed line in **c**, **d** depends on TSL release time. Both in vivo studies and computer simulation demonstrate about ×10 higher efficacy (based on plateau concentration) of fTSL compared to sTSL (Table S1). The right-side y-axis in **c–e** displays the fraction of maximum possible drug uptake ($c_{max}$) achieved. This maximum concentration ($c_{max}$) is equal to the systemic plasma concentration of IV-DDS encapsulated drug. fTSL release was about 8 times as fast as for sTSL (see Table 2).

Table S4, Figs. S13–S14). To obtain a comprehensive picture of how the interaction between physiology, drug and IV-DDS affects delivery efficacy, we performed a parametric study based on ~1200 computer simulations where these two indices were varied. As measure of delivery efficacy, we used the plateau concentration, which represents the highest possible concentration in both plasma and EES achievable for specific index values. Figure 5a shows the results of this parametric study, with the two parameter indices ($R.I.$, $P.I.$) indicated on the horizontal and vertical axes on logarithmic scales. Since the first-pass drug extraction fraction ($EF$) depends directly on the index $P.I.$, $EF$ is indicated as second vertical axis. $EF$ is in the range of ~0.2 to 1 for common chemotherapy agents (see Table S5) and depends both on drug identity and tumor physiology. For our dye and this particular tumor model, we measured $EF = 0.22$ (Table 2), indicated as horizontal dotted line in Fig. 5a. We also calculated the release index $R.I.$ as ratio between tissue transit time and release time ($t_{rel}/TT$) for fTSL and sTSL based on experimental data (Table 2), and indicated this index $R.I.$ for sTSL and fTSL on the horizontal axis (vertical dotted lines). The intersections of the horizontal and vertical dotted lines in Fig. 5a indicate the locations of the experimental results in the parametric map, with the in vivo measured plateau concentration indicated for fTSL and sTSL by the color inside the crosshairs. For example, the location of fTSL in the parametric map indicates that delivery efficacy could be improved by a factor of 2.8 by further reducing the release time, and also suggests that the same fTSL would perform better in tumors with longer transit time than the tumor model used here. Figure 5b indicates how the two indices relate to the rate of drug release from IV-DDS, and the rate of tissue drug uptake.

In addition to the parametric map, we show plasma and EES concentration dynamics for three specific parameter combinations in detail (Fig. 5c). These three cases demonstrate the benefit of a rapidly extracted drug (EF ~ 1), in that less rapid release is required. For less rapidly extracted drugs (e.g., $EF = 0.1$), faster releasing IV-DDS are required for optimal delivery. While the plateau concentration only depends on the two mentioned indices, the specific transport dynamics (e.g., time to reach the plateau in EES) depends on the various other transport parameters; the results presented in Fig. 5c are based on the transport parameters of this specific tumor model (Table 1). The duration required to obtain the plateau concentration in the EES for drugs with very slow extraction ($P.I. < 0.01$, $EF < 0.01$) will be impracticably long (Fig. 5c), and such drugs are not good candidates for the IV-DDS approach due to this limited extraction of released drug.

While the parametric study presented here (Fig. 5a) is based on zero-order IV-DDS release kinetics (Fig. S8), the model can easily be adapted for other IV-DDS release profiles.

## Discussion

Most drug delivery systems (DDS) are based on passive (EPR-based) or affinity targeting for accumulation[8,14,16], but there is a need for alternative approaches due to the inherent limitations of EPR-based delivery[6,18,20]. Triggered drug release represents an exciting concept[12,22–24,27,28,30,51,52] that has been incorporated into passive and affinity targeted DDS. While passive, extravascular DDS accumulation followed by triggered release has been studied extensively[2,8,14,16,30,52–56], the approach of intravascular triggered release has only received more attention in recent years[21,37,57]. The IV-DDS approach is based on the very rapid transvascular transport of free (unencapsulated) drug compared to DDS extravasation (which takes typically 24–48 h[56,58,59]), resulting in rapid drug accumulation in the interstitium (EES), followed by cellular uptake (Fig. 1).

Notably, the transport kinetics of the IV-DDS approach have not yet been fully described or quantified. Mathematical models provide insight into the interplay between biological, drug- and DDS-properties (Fig. S1), and may thus aid the development of more effective DDS. Since transport kinetics for IV-DDS are different from those of free (unencapsulated) drug and from other DDS, standard pharmacokinetic models[49,60] are not applicable. Herein we present an in vivo derived computer model to study IV-DDS and not only describe the phenomenon, but also identify parameters and characteristics to be improved upon in future IV-DDS development. Importantly, all model parameters were directly measured in vivo (Table 1), providing high predictive model accuracy when validated with in vivo studies (Table S1). As IV-DDS model system we employed temperature-sensitive liposomes (TSL), which have been widely studied[21,34–36] and are currently in clinical trials[38,61].

The intravascular triggered delivery approach is based on IV-DDS entering the target tissue volume and releasing drug during transit (Fig. 1), resulting in a longitudinal concentration gradient along the microvasculature (Fig. 3). Once drug is released from IV-DDS, interstitial drug uptake rate depends on transvascular permeability, vascular surface area, and on perfusion. The stimuli-induced intravascular release approach may be conceptualized by an intravascular drug infusion of bioavailable drug simultaneously into all tumor blood vessels (compare Fig. 3a, d to Fig. 3b, e). In fact, hepatic arterial infusion for primary and secondary liver cancer is an effective treatment as long as tumor-feeding arteries are accessible and the drug is well extracted[62,63].

The tissue transit time ($TT$) is an important physiologic property for an IV-DDS. For fast-releasing IV-DDS with complete release within the tissue transit time, the highest possible plasma concentration is achieved ($c_{max}$), and is equal to the systemic plasma concentration of encapsulated drug (Fig. 3b, e). In comparison, incomplete release from IV-DDS results in reduced plasma concentration of free drug (Fig. 3c, f). The importance of this rapid release was directly demonstrated by in vivo studies with two different TSL formulations, with fast and slow release (fTSL and sTSL). The faster fTSL formulation achieved about 10 times higher plateau concentrations in plasma and EES in vivo (Fig. 4c), and the computational model predicted delivery kinetics of both formulations accurately (Fig. 4d, e). Due to localized IV-DDS triggered release, plasma concentration of bioavailable (free) drug is kept elevated within the target region for extended duration, while the trigger is present. Following transvascular drug transport, concentration within the EES equilibrates with plasma concentration within the target region (Fig. 4c, d). Cells are exposed to drug at considerably higher concentrations, and for extended duration compared to standard chemotherapy delivered as a bolus or infusion (compare Fig. 2c, d and Fig. 4c, d), consistent with prior in vivo data[21]. This extended exposure is responsible for the greatly enhanced drug uptake in the targeted tissue compared to administration of unencapsulated drug, with prior TSL studies reporting up to 25-fold enhancement of tumor drug uptake[33]. Since the duration the cells are exposed to the drug directly correlates with the time the trigger is present (Fig. 4c, d), the dose locally delivered to cells can be adjusted based on trigger duration.

As revealed by analysis of the computer model equations, there are two key parameter ratios that exclusively define the maximum achievable concentration (plateau concentration ($c_{plateau}$)) in plasma and EES for a particular IV-DDS, drug, and tumor type: (1) the ratio between IV-DDS release time and tumor segment transit time ($t_{rel}/TT$), and (2) the ratio between vascular permeability-surface area product and plasma perfusion ($PS/F_p$); the latter ratio is tied to the first-pass extraction fraction of free drug ($EF$) (Eq. (1)).

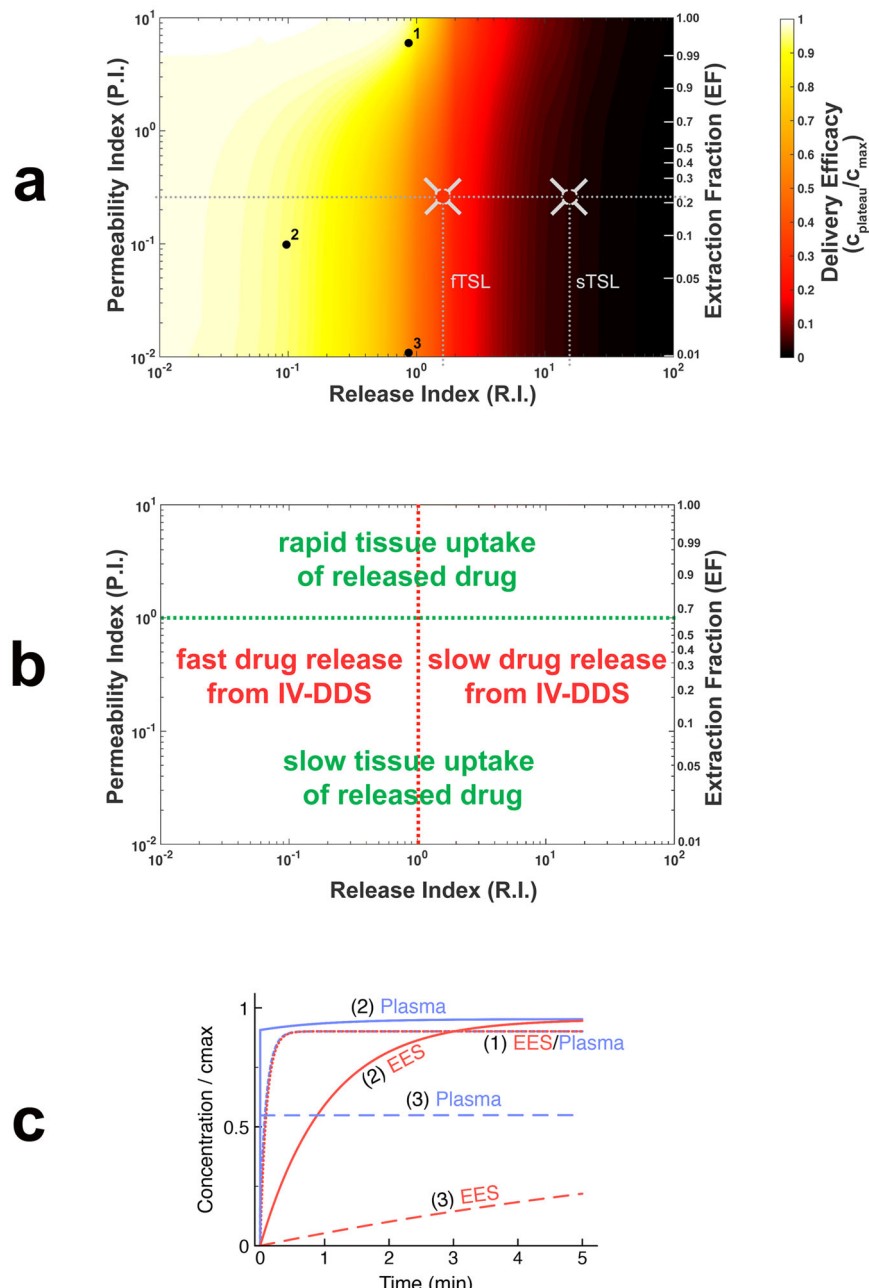

**Fig. 5 Two key parameter indices dictate IV-DDS delivery. a** Computer model was used to conduct a parametric study of the interaction of the two indices that affect the plateau concentration ($c_{plateau}$). These two indices depend on IV-DDS release time ($t_{rel}$), tissue transit time ($TT$), the vascular permeability-surface area product ($PS$), and plasma perfusion ($F_p$): release index $R.I. = t_{rel}/TT$ (x-axis) and permeability index $P.I. = PS/F_p$ (y-axis). First-pass extraction fraction ($EF$) directly depends on $P.I.$ and is indicated on the second y-axis on the right. Delivery efficacy is represented by the plateau concentration relative to maximum ($c_{plateau}/c_{max}$), and is indicated by a color scale. The dotted horizontal line indicates the permeability index $P.I.$ based on in vivo tumor parameter measurements with unencapsulated dye (Table 2); vertical dotted lines indicate the release index $R.I.$ for fTSL and sTSL, based on experimental measurements (Table 2). The intersections of the horizontal and vertical dotted lines indicate the location of experimental in vivo results for fTSL and sTSL in this map, with the color inside the gray crosshairs corresponding to measured in vivo plateau concentration (see Fig. 4c). **b** Parametric map from (**a**) with annotations indicating how release index ($R.I.$) and permeability index ($P.I.$) correspond to rate of IV-DDS drug release and rate of tissue drug uptake, respectively. **c** Concentration time courses for three specific cases, indicated by black dots (marked 1–3) in (**a**) ((**a**) is based on 1200 such simulations). EES and plasma concentration curves are marked (1)–(3) accordingly. The plateau concentration is equal to the concentration at $t = 5$ min for cases (1) and (2); (3) only reaches the plateau at $t \gg 5$ min. Parameters from Table 1 were considered, with exception of $t_{rel}$ and $PS$; the latter were adjusted to obtain desired indices $R.I.$ and $P.I.$. For highly permeable drugs (case (1); EF~1), a release time in the range of the tissue transit time (i.e. $R.I. = t_{rel}/TT <= 1$) is sufficient for near optimal delivery. For lower permeable drugs (case (2), $EF = 0.1$), about 10× faster release is ideal (i.e. $R.I. = t_{rel}/TT <= 0.1$). For drugs with very low permeability (case (3); $EF = 0.01$), the duration to achieve relevant tissue drug uptake is likely prohibitive, making such drugs poor choices. Many common clinical chemotherapy agents have $EF$ in the range of 0.2 to 1 (see Table S5).

**Table 2 In vivo values for key indices.**

| Parameter | Value | Parameter description | Source |
|---|---|---|---|
| $P.I.$ $(=PS/F_p)$ | 0.26 | Permeability Index (=Ratio vascular permeability-surface area product/ tumor plasma perfusion) | Calculated from parameters $PS$, $F_p$ (Table 1) |
| $EF$ | 0.22 | First-pass extraction fraction (Unencapsulated drug) | Calculated from parameters $PS$, $F_p$ (Table 1) |
| $TT$ | $5.0 \pm 0.5$ s | Tissue transit time of tumor segment | Intravital Imaging (Table 1) |
| $t_{rel\_fTSL}$ | 8.2 s | fTSL release time | In vitro essay |
| $t_{rel\_sTSL}$ | 63.0 s | sTSL release time | In vitro essay |
| $R.I._{fTSL}$ $(=t_{rel\_fTSL}/TT)$ | 1.6 | Release Index for fTSL (=Ratio fTSL release time / tissue transit time) | Calculated |
| $R.I._{sTSL}$ $(=t_{rel\_sTSL}/TT)$ | 12.6 | Release Index for sTSL (=Ratio sTSL release time / tissue transit time) | Calculated |

The permeability index $P.I.(=PS/F_p)$ and release index $R.I.(=t_{rel}/TT)$ was calculated from experimental data, and experimental values of the two indices are indicated in Fig. 5a as dotted lines. Column headings are identified in bold font, parameters are identified in italic font.

We defined two indices based on these parameter ratios: the release index $R.I.(=t_{rel}/TT)$, and the permeability index $P.I.$ $(=PS/F_p)$. The impact of these two indices on drug uptake (using plateau concentration ($c_{plateau}$) as measure of delivery efficacy) was illustrated by a parametric study based on the validated computer model (Fig. 5a). The results show that of the two indices, the release index $R.I.$ is more critical. For example, for sTSL ($R.I. = t_{rel}/TT = 12.6$), only 4% of maximum possible drug uptake was achieved, and selecting a drug with more rapid extraction would only provide limited improvement (Fig. 5a). Even the faster fTSL ($R.I. = t_{rel}/TT = 1.6$) didn't utilize the full potential of the IV-DDS approach. In general, a release time less than the transit time is desirable and release times much larger than the transit time ($t_{rel} >> TT$) result in ineffective delivery independent on drug properties (Fig. 5a). Note that release time typically depends not only on the particular IV-DDS but also varies with the encapsulated compound.

While less critical, the permeability index ($P.I. = PS/F_p$)—which is directly tied to the free drug extraction fraction ($EF$) (Eq. (1))—also significantly affects delivery and determines the required release time for optimal delivery. For highly permeable drugs that are completely extracted ($EF \sim 1$), a release time equal or below the transit time is sufficient. For drugs with lower extraction (e.g., for our model drug with $EF \sim 0.2$), a much more rapid release time of less than 10% of transit time (which would correspond to $t_{rel} < 0.5$ s in our tumor model) is ideal (Fig. 5a). Most chemotherapy agents have an extraction fraction ($EF$) in the range of $\sim 0.2$–1 (Table S5), suggesting that the IV-DDS strategy may be widely applied. The typically short initial plasma half-life of chemotherapy agents due to rapid tissue extraction—a shortcoming during conventional chemotherapy—thus becomes an advantage when the drug is combined with IV-DDS. Ideal drugs are thus not necessarily those that are clinically effective in unencapsulated form, and some drugs that failed in clinical trials due to poor biodistribution may be ideal candidates for the IV-DDS approach. Drugs with very slow extraction ($EF << 0.1$) are in general not desirable as drug uptake is very limited (Fig. 5c).

The plateau concentration ($c_{plateau}$) from in vivo studies for fTSL and sTSL (Fig. 4c) is indicated in Fig. 5a by the color inside the gray crosshairs, with locations in the parametric map determined based on in vivo transport parameters (Table 2). This suggests that in addition to aiding IV-DDS design, the parametric map may be employed to predict in vivo performance of an existing IV-DDS based on three in vivo measured transport parameters ($PS$, $F_p$, and $TT$). Notably, two of these parameters (plasma perfusion ($F_p$) and transit time ($TT$)) can be measured non-invasively in human tumors and normal tissues based on dynamic contrast-enhanced imaging[49,64–68]. Thus, the prediction of in vivo performance of a particular IV-DDS in human patients may be possible. Alternatively, an estimate of the extraction

fraction ($EF$) of the encapsulated drug (Table S5) may be used together with measured transit time ($TT$) to predict drug delivery efficacy of an existing IV-DDS based on Fig. 5a.

The two key indices depend on characteristics of tumor physiology (blood perfusion, transit time, and vascular surface area), interactions between drug and physiology (vascular drug permeability), and interactions between drug and DDS (release time) (compare Fig. S1). For a given indication, the properties of the drug and DDS must be properly selected or engineered to achieve the full potential of the IV-DDS. Importantly, these findings highlight the need for fast drug release from IV-DDS and choice of a drug that is well extracted. In addition, physiology (e.g., vascular permeability, perfusion) and biology (e.g., cell uptake) may be modified to further improve delivery via IV-DDS.

Both tumor perfusion and thus transit time (Eq. (2)) are heterogeneous (intra and inter-tumor). The mean transit time of human tumors varies widely, and for example is ~2 s for primary hepatocellular carcinoma[69], ~3 s for head & neck and prostate tumors[65,66], ~11 s for renal cell carcinoma[67], ~25 s for metastases to the liver[69], and ~30 s for breast cancer[68]. Transit time and perfusion vary spatially such that transit time can be locally considerably higher or lower than the mean tumor values listed above. Based on this range of tumor transit times, the required release time ($t_{rel}$) for optimal delivery varies from subsecond range to several seconds (Fig. 5a). Engineering DDS with such rapid release while retaining adequate plasma stability will be one of the challenges in the design of more effective future IV-DDS. An adequate plasma stability of IV-DDS in systemic circulation is necessary to provide an optimal supply of encapsulated drug available for triggered release in the target region, and to limit systemic bioavailability and normal tissue exposure/toxicity.

The presented computational model considers the release trigger (e.g., temperature time course, Fig. 4a) and how IV-DDS release kinetics varies with trigger magnitude (e.g., temperature dependence of TSL release, Figs. S10–S12). Adaptation of the present computer model to other types of IV-DDS would thus require the knowledge of the release trigger dynamics at the target site (e.g., ultrasound intensity for microbubbles, or light fluence rate for light-sensitive IV-DDS), and data on how IV-DDS release kinetics varies with magnitude of the trigger. Since these data only represent input variables to the model, in most cases no changes of the underlying model equations would be required. If there is interaction of the IV-DDS with the microvasculature, the IV-DDS transit time may differ from plasma transit time and would need to be determined independently. In case the 0th order release kinetics is not adequate for a particular IV-DDS (Fig. S8), other release kinetics can be substituted in the model by modifying the release term (see Eq. (10)).

Although not considered in our model, cellular uptake may affect EES drug concentration since cells would act as a drug sink.

Even then, the general relationships presented here are still applicable and we can consider the EES concentration representing the amount of drug available for cell uptake. To include cellular drug uptake in the computer model, an equation would need to be added that describes cell uptake based on the amount of free drug available in the EES. The kinetics of cellular uptake depends on the drug and may also vary by cell type. For common agents such as doxorubicin, platinum drugs (cisplatin, oxaliplatin, carboplatin) and paclitaxel, mathematical models describing cell uptake that were derived from in vitro studies are available[70,71]. Such cell uptake models may be readily integrated in this computational model, similar to prior studies[40,45,56].

**Conclusions**. Various IV-DDS have been studied for decades without guidance on how to engineer an optimal delivery system, and with limited quantitative data on how the interaction between IV-DDS, drug, and tissue impacts delivery. The presented in vivo derived computer model may guide development of improved future IV-DDS by specifying ideal properties of IV-DDS and of drug, and further can predict existing IV-DDS efficacy based on tissue parameters that can be quantified by imaging methods in humans and animals. Since IV-DDS do not rely on passive accumulation via EPR effect, IV-DDS are promising delivery systems not only for cancer treatment, but with numerous other potential applications outside of cancer therapy such as infectious diseases, inflammation, or cardiovascular diseases.

## Methods

**Study design**. The goal of this study was to explain and quantify the delivery kinetics of intravascular triggered drug delivery systems (IV-DDS). For this purpose, we developed methods whereby intravital fluorescence microscopy studies were integrated with a computational model that simulates IV-DDS transport kinetics (Fig. 1). The purpose of the intravital studies was (1) to quantify all tumor transport parameters necessary for the computational model, and (2) for direct validation of drug concentration dynamics in tumor plasma and interstitium (extravascular-extracellular space (EES)) predicted by the computational model. We used the fluorescent dye carboxyfluorescein as drug analog. This dye has negligible cell uptake[72], which was required (1) to quantify transvascular transport parameters (cell uptake represents an additional sink, and would thus change the rate of transvascular transport), and (2) for the direct conversion of fluorescence to absolute concentration (after cellular uptake, interaction with intracellular components changes the magnitude and spectrum of fluorescence).

In a first set of intravital experiments, tumor transport parameters were quantified after bolus administration of unencapsulated dye during continuous imaging, followed by quantitative image processing. Pharmacokinetic (PK) parameters were quantified based on serial blood sampling following dye administration in additional animal studies. A computational drug delivery model was developed based on these in vivo parameters, and the ability to reproduce delivery kinetics of unencapsulated drug was validated based on this first set of intravital experiments (Fig. 6).

For validation of the computer model simulating IV-DDS delivery, a second set of intravital studies was performed where thermosensitive liposomes (TSL) - which release drug triggered by hyperthermia - were used as example IV-DDS. We performed intravital studies with two TSL formulations (with fast and slow release, Fig. S10) and performed quantitative image analysis (Fig. S3) to extract the temporal dynamics of intra- and extravascular dye concentration. We used these data for direct validation of the computer model predictions (Fig. 7).

Finally, the validated computer model was employed to explain and quantify IV-DDS delivery kinetics. The key transport parameters that determine IV-DDS based drug delivery were identified, and we established how the interaction of these key parameters dictates tumor drug uptake.

**Definition of parameters**. Below we define and describe parameters that are used throughout the manuscript, and referenced below. We used a dye as drug analog in this study, and any reference to "drug" is also applicable to drug analogs as employed here. The parameters are indicated within the appropriate tissue compartment in Fig. S9. Note that some of the parameters below apply specifically to the imaging window and may be different from mean tumor values.

**Vascular volume fraction**, $v_v$: Fraction of tumor tissue volume contained by vasculature. Note however that the vascular density in the imaging window is higher and not representative of the whole tumor—in part due to overestimation of vascular volume based on 2D intravital image analysis[73].

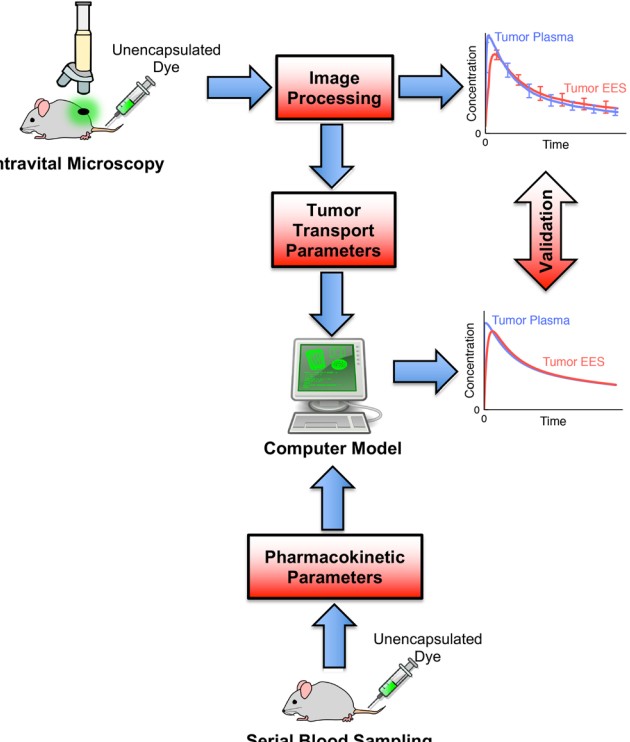

**Fig. 6 Measurement of transport parameters and computer model validation with unencapsulated dye.** Based on intravital studies with unencapsulated dye, tumor transport parameters were quantified. Pharmacokinetic parameters were measured from additional animal studies with serial blood sampling following dye administration. A computer model based on these parameters simulated drug delivery kinetics of unencapsulated dye, and computer model results were validated by intravital imaging studies (right-side graphs).

**Plasma volume fraction**, $v_p$: Fraction of tumor tissue volume contained by plasma. $v_p = v_v*(1−HcT^{MV})$; $HcT^{MV}$ is the hematocrit inside tumor microvasculature, which is lower than systemic hematocrit ($HcT^{MV} = 0.19$)[74].

**Extravascular volume fraction**, $v_e$: Fraction of tumor tissue volume in imaging window contained by extravascular space.

**Available fraction of extravascular volume**, $k_{av}$: Fraction of extravascular space in imaging window that is available to drug or drug analog (i.e., dye) for distribution.

**Available volume fraction**, $v_{e,av} = k_{av} * v_e$: Fraction of volume available (i.e., fraction of total volume) for drug or drug analog (i.e., dye) distribution, calculated by multiplying the extravascular volume fraction ($v_e$) with the fraction of this volume that is available for dye distribution ($k_{av}$).

**Vascular drug permeability**, $P$ [cm/sec]: Permeability of vascular wall towards unencapsulated drug. In our study, this is not the true vascular permeability but represents an apparent permeability that implicitly may represent diffusion as well as convective transport mechanisms[47].

**Vascular surface area**, $S$ [mm²/mL]: Vascular surface area per tissue volume.

**Vascular permeability-surface are product**, $PS$ [1/s]: The product of vascular drug permeability and surface area. As is often done, the two parameters P and S are combined into a single parameter since the product determines tissue uptake rate of drug. Furthermore, it is often easier to measure this product than to measure $P$ and $S$ individually.

**Plasma perfusion rate**, $F_p$ [mL/(mL*s)]: Tumor perfusion in mL of plasma per mL of tumor tissue per second.

**Tissue transit time**, $TT$ [s]: Average time that plasma requires to pass through a tumor tissue segment.

**IV-DDS release time**, $t_{rel}$ [s]: Time required for complete release of encapsulated drug from IV-DDS (e.g., thermosensitive liposome). This time is based on a constant release rate (zero-order release kinetics, see Fig. S8).

**First-pass extraction fraction**, $EF$: Fraction of unencapsulated drug extracted by tissue during the first pass through the tissues microvasculature (Fig. 3d). $EF$ depends on the ratio of $PS/F_p$ (Eq. (6)).

**Systemic plasma volume**, $V_p^S$ [mL]: Total volume of plasma in systemic circulation.

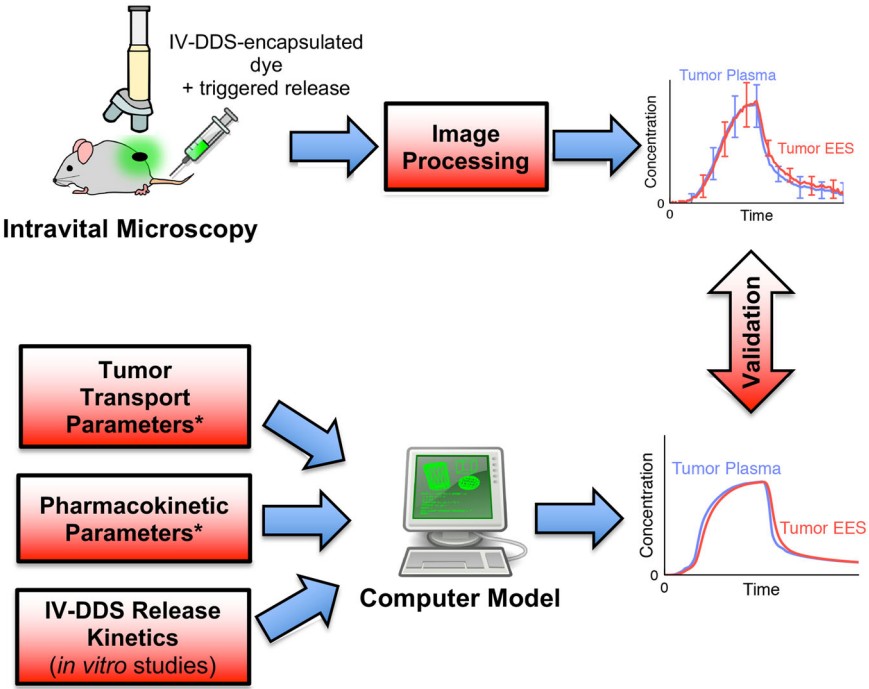

**Fig. 7 Validation of IV-DDS computer model by quantitative intravital microscopy.** A computer model was developed to simulate delivery kinetics of IV-DDS. The model was validated by quantitative intravital microscopy studies where fluorescent dye encapsulated in thermosensitive liposomes (TSL) served as example IV-DDS. The computer model was based on in vitro measured IV-DDS (i.e., TSL) release kinetics, and on in vivo measured parameters from the in vivo studies with free dye (parameters marked by asterisks (*), see Fig. 6). The computer model predicted intra- and interstitial/EES concentrations during IV-DDS delivery, which were validated against results from the intravital studies (right-side graphs).

**Initial volume of distribution**, $V_D$ [mL]: Distribution volume of unencapsulated drug (here measured 2 min after bolus administration).

**Transport rate constants** $k_p$, $k_t$, $k_e$ [1/s]: Rate constants describing pharmacokinetics of unencapsulated drug based on a two-compartment pharmacokinetic model (Fig. S4).

**Thermosensitive liposome (TSL) preparation**. Liposomes were made using the film hydration technique using different ratios of lipids: DPPC:DSPC:DSPE-PEG2000 in a ratio of 80:15:5 for fast (fTSL) or in a ratio of 55:40:5 for slow (sTSL) release of carboxyfluorescein (CF). Lipids were dissolved in chloroform:methanol [9:1 (v/v)]. Organic solvents were evaporated using a rotary evaporator (Büchi-R144, Switzerland) at 40 °C to obtain a lipid film, which was dried under vacuum. The film was hydrated with 4 mL of 100 mM CF solution (pH 7.2) at 60 °C and extruded through 200, 100, 80, and 50 nm polycarbonate membranes to obtain a homogeneous liposomal population. Unencapsulated CF was eliminated by using a PD10 column with Hepes saline buffer (Hepes 10 mM, NaCl 150 mM and, ethylenediaminetetraacetic acid (EDTA) 5 mM, pH 6.7). Liposomes were stable in 90% serum (either fetal calve serum (FCS), mouse serum, or human pooled serum) up to 37 °C. Liposomes were stored at 4 °C until use. Size, polydispersity index, and zeta potential of sTSL and fTSL were determined by dynamic light scattering using a Zetasizer Nano ZS (Malvern Instruments, Worcestershire, UK) (Table S6).

**TSL in vitro release characterization**. Time dependent release profiles of CF, i.e. speed of CF release at temperatures of 38–42 °C, from both sTSL and fTSL was determined as described earlier[37]. A liposome stock solution of 5 mM was prepared in HEPES buffer and 50 µL liposomes was added to 2950 µL FCS in a quartz cuvette that was preheated to the desired temperature of 37, 38, 39, 40, 41, or 42 °C. The temperature in the cuvette was monitored with a thermocouple thermometer during the assay. The sample was continuously stirred, and fluorescence ($Fl_n$) recorded every second for 1 h ($\lambda_{ex}$ = 492 nm; $\lambda_{em}$ = 517 nm). After 1 h, 50 µL Triton-X100 was added to the sample to obtain 100% release ($Fl_m$). Background fluorescence ($Fl_b$) was recorded before addition of liposomes. Percentage release for every time point was calculated according to:

$$\text{Release (\%)} = (Fl_n - Fl_b)/(Fl_m - Fl_b) * 100\%$$

**Dorsal skin fold window chamber**. All animal studies described below were approved by the institutional animal care and use committee of the Erasmus Medical Center (approval nr. AVD101002016792), and the experiments were performed according to the European directive 2010/63/eu on the protection of animals used for scientific purposes. Male C57BL/6 mice (n = 9), ~12 weeks old, were anesthetized (inhalation of isoflurane) and hair was removed from the back of the animal. After dissecting the skin on one side, leaving the fascia and opposing skin, the skin-fold of the mouse was sandwiched between two frames, fixed with two light metal bolts and sutures[75]. Murine Lewis Lung Carcinoma (LLC) cells were cultured in DMEM medium (Lonza, Belgium) containing 10% (v/v) FCS, and incubated at 37 °C in a humidified environment of 95% air and 5% $CO_2$. LLC cells were tested for mycoplasma contamination. $10^6$ tumor cells were injected subcutaneously in flanks of mice, and after the tumor reached ~10 mm³ volume, a small tumor piece (2 mm³) was removed and transplanted in the fascia of the dorsal skin flap window chamber. The chamber was closed with a 12 mm cover glass and the tumor allowed to grow for 1 week. The mice were housed in an incubation room with an ambient temperature of 32 °C and a humidity of 70%.

*Intravital microscopy studies*. One week after tumor implantation, mice (n = 9) with window chamber and tumor were assigned randomly to receive an injection of either unencapsulated carboxyfluorescein (CF) dye, or one of two TSL formulations. Tumors were imaged using intravital confocal microscopy (Zeiss LSM 510META confocal microscope) under anesthesia[75]. During imaging, mouse body temperature was kept stable at 37 °C using a thermal stage and monitored by a rectal probe. The temperature in the window was continuously monitored using a thermocouple inserted into the window chamber.

*Unencapsulated (free) dye studies*. Unencapsulated CF dye (8 µmol/kg) was administered as bolus in 100 µL phosphate-buffered saline (PBS) intravenously to tumor-bearing mice (n = 3). The window was kept at 37 °C using an external circular electric heating coil that was attached to the window glass on the backside of the window chamber. Images were taken every 4 s starting before bolus administration of dye, and up to 20 min after injection.

*TSL-encapsulated dye studies*. Studies were performed in C57BL/6 mice with two TSL formulations (sTSL and fTSL described above, n = 3 each) encapsulating CF dye. At given time points after tumor implantation, mice were anesthetized and fixed to the heated microscope stage of a Zeiss LSM 510META confocal microscope. After bolus injection, the tumor was heated by an external circular resistive electric heating coil attached to the glass at the backside of the window chamber to 42 °C for 10 min (Fig. 4a). Continuous imaging (every 10 s) was performed starting before initiation of heating, and up to 10 min after heating was stopped. CF dye release was detected by a 488 nm argon laser and emission filter set of 500–550 nm, and images were acquired at three gain settings to ensure at least one image data set

of adequate quality (i.e. a signal of sufficient magnitude but without saturation) was obtained.

**Quantitative image processing**. In most image datasets there was significant motion, and sometimes deformation was present. Motion and deformation compensation of imaging data was performed based on custom scripts (MevisLab 2.7.1, Fraunhofer MeVis) to perform image registration between each image frame and a reference frame (i.e., each image frame was transformed (translation, rotation, and scaling) to match the reference frame optimally). To compare an image frame to the reference frame, the normalized mutual information (NMI) was used to quantify the similarity between images[76]. Through an iterative procedure, a certain image frame was transformed using affine transformation which can correct for rotation, translation, and scaling. Transformation parameters were iteratively adjusted according to the conjugate gradient method until an optimal match between the reference image and transformed image frame was achieved[76]. This iterative registration procedure was performed for each image frame. This produced datasets without motion or deformation artifacts, where a static vascular mask could be applied to obtain mean intra- and extravascular fluorescence of the tissue segment over time. Background fluorescence (before administration of liposomes or dye) was subtracted. Intra- and extravascular masks were defined based on images showing only the quenched fluorescence of TSL-encapsulated dye (since TSL remain within the vasculature, see Fig. S3a), i.e. before any release. For free CF studies, an initial time frame where extravasation could be identified was used to define vascular and extravascular masks (Fig. 2a). Based on these masks, the time course of mean intra- and extravascular CF intensity ($I_v(t)$, $I_e(t)$) was calculated (Fig. S3c). These data were used to quantify transport parameters as described below.

*Conversion of intravital fluorescence to concentration*. Intra- and extravascular fluorescence intensity data (Fig. S3c) were converted to absolute concentration based on calibration data from plasma samples where dye concentration was quantified (Fig. S2).

Unencapsulated dye studies: Quantitative measurements of plasma concentration of dye were performed in "Dye clearance studies to determine pharmacokinetic parameters" described below. These data were employed to calibrate intravascular fluorescence, and convert to a plasma concentration. Extravascular fluorescence was converted to concentration considering the available volume fraction in extravascular space ($k_{av}$) determined as part of the "Quantification of tumor transport parameters" described below.

Thermosensitive liposome (TSL) studies: Plasma concentration of encapsulated and unencapsulated CF was quantified from plasma samples 1 min after administration of both sTSL and fTSL ($n = 3$ per group). We found negligible amount (<1%) of unencapsulated dye present in plasma for both TSL formulations. Thus, plasma fluorescence before heating (i.e. release) corresponds to fluorescence of encapsulated CF, which is quenched due to encapsulation[72]. Based on in vitro studies, we measured that fluorescence increases by a factor of 30.2 after complete release from TSL, compared to before release. Based on these data, we calculated the concentration of unencapsulated (i.e., released) drug present in tumor plasma during hyperthermia from the intravital imaging data, using intravascular fluorescence of encapsulated CF before hyperthermia as reference. Extravascular fluorescence was again converted to concentration based on the available volume fraction in extravascular space ($k_{av}$).

**Quantification of tumor transport parameters**. The analyses below were performed based on the intravital imaging data from the studies with bolus administration of unencapsulated dye (Figs. 6, 2).

*Plasma and extravascular volume fractions*. Based on the vascular mask (Fig. S3b), the fractions represented by vascular ($v_v$) and extravascular tissue volume ($v_e$) were determined considering the fraction of the image represented by vascular and extravascular space, respectively. Plasma volume fraction ($v_p$) was then determined from the vascular fraction ($v_v$) considering the tumor microvascular hematocrit: $v_p = v_v*(1-Hct^{MV})$, where $Hct^{MV} = 0.19$[74]. Note that this plasma volume fraction $v_p$ only applies to the imaging window where tumor microvasculature was imaged. Therefore, we additionally assume a separate plasma volume fraction applicable to the whole tumor in the model where systemic hematocrit was considered (see Table S2).

*Vascular permeability analysis*. We used an approach derived from methods presented in a prior publication[47]. Based on a 2-compartment model (Eq. (3)), an equation was derived (Eq. (4)) to describe the relationship between intravascular and extravascular fluorescence intensity (rather than concentration) measured in intravital studies. The derivation of Eq. (4) required conversion of concentration to fluorescence, which was done based on the available volume fractions in intra- and extravascular space. The available volume fraction in extravascular space is $k_{av}$. In intravascular space, the available volume fraction is equal to the blood plasma fraction and therefore determined by the hematocrit in the tumor microvasculature ($Hct^{MV} = 0.19$[74]). Considering these available volume fractions, Eq. (4) was

derived from Eq. (3):

$$\frac{dc_e}{dt} = \frac{1}{v_e} PS \left( c_p(t) - c_e(t) \right) \tag{3}$$

$$\frac{dI_e}{dt} = \frac{1}{v_e} PS \left( \frac{I_v(t)}{(1 - Hct^{MV})} - \frac{I_e(t)}{k_{av}} \right) \tag{4}$$

The extravascular volume fraction ($v_e$) has been already quantified as described above. Thus, Eq. (4) has two unknown parameters: the permeability-surface area product ($PS$), and the available volume fraction in extravascular space ($k_{av}$). Intra- and extravascular fluorescence signals ($I_v(t)$, $I_e(t)$) were extracted from intravital image data following bolus administration of free dye (Fig. 2c). Based on these fluorescence signals ($I_v(t)$, $I_e(t)$) an unconstrained nonlinear optimization was performed in the software Matlab 2020a to determine the two unknown parameters ($PS$, $k_{av}$).

*Tissue transit time analysis*. This analysis was performed based on the animal studies with bolus injection of free dye, considering the data acquired during the first 30 s following injection (Fig. S15). The microvascular tissue transit time ($TT$) describes how long on average plasma (and therefore a nanoparticle, e.g., liposome) spends within the microvasculature of a tissue region (note that transit time of red blood cells is often longer as they interact with and squeeze through microvasculature). Dynamic contrast-enhanced (DCE) imaging methods such as DCE-MRI commonly calculate a mean transit time within a tissue region based on bolus administration of a contrast agent[49], and here we used an equivalent approach. The transit time can be calculated from the deconvolution of the intravascular fluorescence signal $I_v(t)$ (i.e. the fluorescence averaged within the vasculature based on a vascular mask of the whole image) (Fig. S15), and an arterial input function $AIF(t)$ that corresponds to the intravascular signal at the tumor feeding artery. As we performed a rapid bolus injection, we could assume a step function as input function ($AIF(t)$); we confirmed the validity of this assumption by calculating the fluorescence signal at the entrance vessels segments (i.e., the first vessel segments that become visible). We performed the deconvolution based on an optimization-based deconvolution algorithm described in a prior study, assuming exponential washout[77].

Based on transit time ($TT$), Plasma perfusion ($F_p$) was then calculated according to:

$$F_p = \frac{v_p}{TT} \tag{5}$$

The first pass extraction fraction ($EF$) of unencapsulated drug/dye represents the fraction extracted by tissue during the first pass through the tissue segments microvasculature. $EF$ was calculated from the permeability-surface area product ($PS$) and from plasma perfusion ($F_p$) according to[49]:

$$EF = 1 - e^{-PS/F_p} \tag{6}$$

**Dye clearance studies to determine pharmacokinetic parameters**. C57BL/6 mice were injected with different doses of free CF: 3, 6, 12, 48, and 60 μmol/kg ($n = 6$ at each dose). Blood aliquots were obtained at 2, 4, 8, 16, 32, 64, and 128 min ($n = 3$ at each time point) after injection in heparine-EDTA tubes. Blood samples were centrifuged at 3500 rpm for 15 min at 4 °C and plasma samples were kept at −20 °C until analysis. CF levels in plasma were quantified in black 96 well plates using a standard curve of CF (dilution starting at 0.1 mM) in Hepes buffer in triplicate. Non-treated mouse plasma was measured for background fluorescence correction. The fluorescence measurements were performed in a Wallac VICTOR2 plate reader ($\lambda_{ex} = 492$ nm; $\lambda_{em} = 517$ nm). The initial volume of distribution ($V_D$) was then calculated from the first plasma concentration time point at 2 min, by dividing plasma concentration and injected dose. Based on a two-compartment model (plasma and tissue compartment) (Fig. S4), a bi-exponential fit was performed to the plasma concentration data (Fig. S7):

$$C(t) = C_0 \left( \alpha \cdot e^{-\lambda_1 t} + (1 - \alpha) \cdot e^{-\lambda_2 t} \right) \tag{7}$$

From this bi-exponential fit, the three rate constants for elimination ($k_e$), and tissue distribution ($k_p$, $k_t$) were calculated:

$$k_t = \frac{\alpha(1-\alpha)(\lambda_2 - \lambda_1)^2}{(1-\alpha)\lambda_1 + \alpha\lambda_2} \quad k_p = (1-\alpha)\lambda_1 + \alpha\lambda_2 \quad k_e = \frac{\lambda_2\lambda_1}{(1-\alpha)\lambda_1 + \alpha\lambda_2} \tag{8}$$

*Linearity of in vivo fluorescence*. Since we performed a conversion of fluorescence to absolute concentration, we also confirmed the linearity between fluorescence and concentration. For this purpose, we compared dye concentration based on quantification of serial blood sampling (described above) to intravascular fluorescence at the same time points. We used the averaged data from the serial blood sampling studies and compared time points before administration, as well as after 2, 4, 8, and 16 min to fluorescence imaging studies (note, imaging studies were limited to 20 min). The results demonstrate excellent in vivo linearity (Fig. S2).

**Statistics and reproducibility**. Mice were pooled and selected randomly for experiments. The produced liposomal formulations were used randomly after quality control based on the described in vitro characterization methods. In vivo

---

**Table 3 Variables used in computational model.**

| Variable [Units] | Variable description |
|---|---|
| $c_p^T(x,t)$ [µg/ml] | Tumor plasma concentration of unencapsulated drug along microvasculature |
| $c_{p\_DDS}^T(x,t)$ [µg/ml] | Tumor plasma concentration of IV-DDS encapsulated drug along microvasculature |
| $c_e^T(t)$ [µg/ml] | Tumor interstitial/EES concentration of unencapsulated drug |
| $c_{p,DDS}^S(t)$ [µg/ml] | Systemic plasma concentration of IV-DDS encapsulated drug |
| $c_p^S(t)$ [µg/ml] | Systemic plasma concentration of unencapsulated drug |
| $c_t^S(t)$ [µg/ml] | Systemic tissue concentration of unencapsulated drug |
| $x$ [1] | Normalized spatial dimension along capillary, range [0...1] |
| $t$ [s] | Absolute time |

The variables represent the various concentrations solved in the computer model, and are indicated in the appropriate compartments in Fig. S9 showing a schematic of the computational model. Superscript 'T' in variable names indicates tumor compartments, and superscript 'S' indicates systemic compartments. Subscript 'p' indicates plasma, subscript 'e' indicates EES/interstitium, and subscript 't' indicates tissue.
Column headings are identified in bold font, variables are identified in italic font.

---

studies and in vitro liposome characterizations were performed at the Erasmus Medical Center without blinding. All data were transferred to the Medical Univ. of South Carolina, where image processing and data analyses were performed. All animal studies were replicated in separate animals ($n = 3$), which provides a 90% confidence interval of ±0.95 standard deviations. Since the purpose of the in vivo studies was validation of the computer models, no statistical comparisons between individual groups in animal studies were performed.

**Computational drug delivery model.** The computational model consisted of two coupled component models: (1) a vascular gradient model that simulated the concentration gradient of IV-DDS encapsulated drug and unencapsulated drug along the tumor microvasculature; and (2) a multi-compartmental model that simulated drug transport between tumor plasma and tumor interstitium(EES) and includes systemic plasma and -tissue compartments (see Fig. S9). Concentration of unencapsulated (free), as well as encapsulated drug, was modeled in plasma. The modeled variables are listed in Table 3, and all model parameters are listed in Table S2. Both the variables and parameters are indicated in the appropriate compartments in Fig. S9 that shows the overview schematics of the computer model.

*Vascular gradient model.* This vascular gradient model was used to calculate the concentration of a drug (or drug analog, e.g., fluorescent dye as used here) along the microvascular capillaries (along $x$), i.e., between the supplying artery and draining vein of a tissue segment (see Fig. 3). Concentration of unencapsulated (free), as well as encapsulated drug in plasma, was modeled. We assumed that there is rapid equilibration in extravascular (EES) concentration, and thus consider EES concentration independent of $x$. We further assumed that IV-DDS release their contents at constant rate (i.e., zero-order release kinetics), as already discussed above (Fig. S8).

**Vascular gradient model equations:**
Tumor plasma concentration gradient (free drug):

$$\frac{dc_p^T(x)}{dx} = \frac{PS}{F_p}(c_e^T(t) - c_p^T(x)) + \frac{TT}{t_{rel}} \cdot c_{p\_DDS}^S(t) \quad \left[c_{p\_DDS}^T(x) > 0\right]$$
$$\frac{dc_p^T(x)}{dx} = \frac{PS}{F_p}(c_e^T(t) - c_p^T(x)) \quad \left[c_{p\_DDS}^T(x) = 0\right]$$
(9)

The first right-hand term in Eq. (9) represents the amount of drug extracted along the capillary. Extravascular concentration was assumed constant without dependence on distance $x$, i.e., rapid equilibration of concentration within extravascular space was considered. The extravascular concentration $c_e^T(t)$ and systemic concentration of encapsulated drug $c_{p\_DDS}^S(t)$ were calculated via the compartment model described below, i.e., the two models were coupled. The second right-hand term represents drug release from IV-DDS considering zero-order release kinetics, and this term is zero once all drug has been released as indicated in second Eq. (9).
Tumor plasma concentration gradient (IV-DDS encapsulated drug):

$$\frac{dc_{p\_DDS}^T(x)}{dx} = -\frac{TT}{t_{rel}} \cdot c_{p\_DDS}^S(t) \quad [c_{p\_DDS}^T(x) > 0]$$
$$\frac{dc_{p\_DDS}^T(x)}{dx} = 0 \quad [c_{p\_DDS}^T(x) = 0]$$
(10)

The ratio $TT/t_{rel}$ in the first Eq. (10) represents a normalized IV-DDS drug release rate. The second Eq. (10) represents a condition to ensure that IV-DDS concentration does not assume negative values (only relevant if $t_{rel} < TT$, i.e., for IV-DDS with complete release during transit). The variable $c_{p\_DDS}^S(t)$ represents the systemic plasma concentration of encapsulated drug entering the tumor vasculature. IV-DDS release kinetics different from the zero-order kinetics assumed here can be implemented by modifying Eq. (10) accordingly. For example, by assuming a varying release rate depending on $x$ (i.e., $TT/t_{rel}$ is replaced by an

arbitrary function $R = f(x)$), any desired release kinetics profile may be implemented.
Cumulative drug amount extracted by tissue up to location $x$:

$$c_{extr}^T(x) = \frac{PS}{F_p} \int_{\xi=0}^{x} (c_p^T(\xi) - c_e^T)d\xi$$
(11)

The total drug amount extracted by tissue is:
$c_{extr}^T \ (x = 1)$
While in the equations above we use distance $x$ as variable, this distance is equivalent to the residence time of plasma within the microvessel segment. To avoid confusion with absolute time $t$ (as used below in the compartmental model equations), we elected to use distance $x$ for the microvascular gradient equations rather than time $t$. Since the equations above are evaluated at each time step of the compartmental model solution, each of the concentration variables above is also dependent on absolute time $t$, even though this is not indicated in the equations.
Boundary conditions:

$$c_p^T(x = o) = c_p^B$$

$$c_{p\_DDS}(x = 0) = c_{p\_DDS}^s$$

Unencapsulated drug was simulated via the same equations, except that any equations and boundary conditions describing IV-DDS were not considered.

*Compartmental model.* Systemic plasma, systemic bodily tissues, and extravascular target tissue volume (i.e., tumor) were each represented individually by perfectly mixing compartments (Fig. S9). A difference to prior models is that tumor microvasculature was not considered perfectly mixed, but that we considered a concentration gradient (for both encapsulated and unencapsulated drug) along the microvasculature as determined by the gradient model described above. The gradient model was executed after each time step from within the compartmental model to update the microvascular concentration gradients for free and encapsulated drug. Exchange between tumor plasma volume and systemic plasma takes place via plasma perfusion ($F_p$). The concentration of encapsulated drug within the systemic plasma ($c_{p\_DDS}^S$) is reduced due to drug released from the IV-DDS. A bolus administration of IV-DDS at time $t = 0$ was assumed, with drug release occurring only in the target region once triggered release was initiated. The volume of distribution for IV-DDS was considered equal to the systemic plasma volume of the animal[45,78], as liposomes remain within the vasculature for extended duration following administration. The systemic mouse plasma volume was calculated based on body weight[79]. The IV-DDS release kinetics are dictated by the release time ($t_{rel}$), and we assumed zero-order release (i.e. release at a constant rate; Fig. S8). IV-DDS enter the tumor vasculature where triggered release occurs and we assumed release stops once the IV-DDS exit the target volume via draining vein, since the trigger is no longer present. The transport of bioavailable (or free) drug (i.e. after release) from plasma space into EES depends on the permeability-surface area product ($PS$) as well as the difference between plasma- and EES concentrations, and was determined as part of the gradient model (Eq. (9)). The amount of drug extracted by tumor tissue was also determined based on the vascular gradient model (Eq. (11)). While this transvascular transport can include both diffusion and convection mechanisms, we considered an apparent permeability constant $P$ that may represent various transport mechanisms[45,47,78]. Drug that is not extracted by the target tissue is carried away by perfusion back into systemic plasma. This free drug can subsequently be taken up by systemic tissues, and is also removed via clearance. Extravasation of IV-DDS into tumor interstitium (EES) was assumed negligible. Due to the substantially larger size and lower vascular permeability of IV-DDS compared to free drug, IV-DDS extravasation typically takes 24–48 h, i.e. orders of magnitude longer than the duration considered here[45,53,78].

**Compartment model equations:**
The compartments modeled include tumor interstitium (EES), systemic plasma, and systemic tissue.

Tumor EES concentration (free drug):

$$\frac{dc_e^T(t)}{dt} = \frac{1}{v_e} F_p \cdot (c_{extr}^T (x=1)) \tag{12}$$

Systemic plasma concentration (free drug):

$$\frac{dc_p^S(t)}{dt} = \frac{V_p^T}{V_p^S} \frac{F_p}{v_p} \left( c_p^T(x=1) - c_p^S(t) \right) f_{distr} - k_e c_p^S(t) - k_p c_p^S(t) + k_t c_t^S(t)$$

$$f_{distr} = 1 \left[ \left( c_p^T(x=1) - c_p^S(t) \right) \le 0 \right] \tag{13}$$

$$f_{distr} = \frac{V_p^T}{V_D} \left[ \left( c_p^T(x=1) - c_p^S(t) \right) > 0 \right]$$

The factor $f_{distr}$ is employed to apply the initial distribution volume $V_D$ for any drug that is released into tumor plasma (e.g., by release from IV-DDS, or back-diffusion from EES into tumor plasma). This initial distribution volume $V_D$ models the rapid initial distribution of drug after entering the systemic circulation, compared to the slower uptake and distribution modeled by the rate constants $k_t$ and $k_p$.

Systemic tissue concentration (free drug):

$$\frac{dc_t^S}{dt} = k_p c_p^S - k_t c_t^S \tag{14}$$

Systemic plasma concentration of IV-DDS encapsulated drug:

$$\frac{dc_{p\_DDS}^S(t)}{dt} = \frac{V_p^T}{V_p^S} \frac{F_p}{v_p} (c_{p\_DDS}^T(x=1) - c_{p\_DDS}^S(t)) \tag{15}$$

The relationship between blood perfusion $F$[mL blood/mL tissue/s], plasma perfusion $F_p$[mL plasma/mL tissue/s] and transit time $TT$[s] is:

$$\frac{F_p}{v_p} = \frac{F}{v_v} = \frac{1}{TT} \tag{16}$$

Initial conditions: The initial concentrations ($t=0$) for simulation of IV-DDS encapsulated drug based on bolus administration of a specific injected dose ($ID$) were:
$c_p^S = c_t^S = c_p^T = c_e^T = 0$
$c_{p\_DDS} = ID/V_p^S$.
For simulation of unencapsulated drug, the initial conditions were:
$c_t^S = c_p^T = c_e^T = 0$
$c_p^S = ID / V_D$

*Model implementation.* The ordinary differential equations above were implemented in the software Matlab 2020a, and were solved via a single-step solver based on an explicit Runge-Kutta (4,5) formula[80].

*Computer model assumptions.*

- *No systemic leakage of drug from IV-DDS during triggered release* (~10 min in our case). I.e., the IV-DDS have adequate plasma stability. As discussed above, we confirmed adequate plasma stability of the TSL used as model system here at body temperature (Fig. S10b).
- *Extravascular binding of dye is negligible.* This assumption is adequate, as demonstrated by the equilibration between plasma and EES concentrations following tissue uptake (Fig. 2c), and by the ability of the computer model to simulate the delivery kinetics of unencapsulated dye without considering extravascular binding (Fig. 2d, e).
- *Linear impact of any light absorption by tissue components.* Light absorption by tissue components may reduce fluorescence. As long as such a reduction is linear (i.e. reduction factor independent on concentration), this effect is factored into the available volume fraction $k_{av}$ (i.e. $k_{av}$ is lower than the true value), but does not cause any error in concentration values.
- *Plasma protein binding of dye.* While not considered explicitly in the model equations, any plasma protein binding would result in a reduced apparent vascular permeability. Since we measured the vascular permeability-surface area product in vivo, any such binding is considered implicitly in the apparent vascular permeability.
- *Transvascular transport can be modeled as diffusion.* While transport processes other than diffusion could potentially contribute to transvascular transport (e.g., convection), the modeling approach assuming diffusion transport is appropriate as long as net transport can be represented accurately by an apparent permeability, and this method has been employed in numerous prior studies[45,47,78]. Furthermore, for small molecules such chemotherapeutic drugs, diffusive transport is typically dominating over convection due to the comparably high diffusivity of such agents[43].
- *Cell uptake is negligible.* The dye employed here has negligible cell uptake, as demonstrated in earlier studies[72].
- *Release volume small compared to systemic tissue volume.* In in vivo studies and in the presented computer model results, the ratio between tumor plasma volume (where release is triggered from IV-DDS) and

systemic plasma volume was ~1:700. As a result, only 10.6% of the administered IV-DDS released the contained drug during heating. Any recirculation of IV-DDS that already passed the tumor once and released dye during a second pass at a different rate has negligible impact for the conditions of our in vivo experiments. To confirm this assumption, we performed additional computer modeling studies where multiple passes of IV-DDS through the tumor were considered (Suppl. Note 2, Tables S7–S8, Fig. S16, Eq. S1–S3).

*Release time calculation.* In the computer model we assumed zero-order release kinetics, i.e., a constant release rate (Fig. S8, Eq. (10)). While release rate is not constant for sTSL or fTSL for longer time scales of minutes (Fig. S10), for intravascular triggered release only the release within the transit time is of relevance (Fig. S8); after passing through the tissue segment where release is triggered, the IV-DDS (sTSL or fTSL) returns to systemic circulation without further release. For such short time scales, both sTSL and fTSL do exhibit zero-order release kinetics (Figs. S11, S12). Based on a linear approximation of the data, IV-DDS release time ($t_{rel}$) was calculated for each temperature (Fig. S8, Table S3). Since sTSL revealed linear release for longer than fTSL, and because of the much lower amount released (<1 % at most temperatures), we used data during the first 10 s to calculate release time ($t_{rel}$) for sTSL (Fig. S12), and during the first 4 s for fTSL (Fig. S11). As the calculated release time is based on the first 4 or 10 sec, this time does however not indicate the true time to complete release (Fig. S8).

*Simulation of unencapsulated drug.* We simulated delivery kinetics of unencapsulated drug assuming bolus injection into systemic plasma at $t=0$, with immediate distribution within systemic plasma. The intra- and extravascular concentration was calculated over 20 min and compared to results from intravital microscopy studies. We calculated the mean absolute error between experimental (indicated by overline on top of variable $c_e^T$) and predicted extravascular concentrations during the first 20 min ($=T$) according to:

$$MAE = \frac{1}{T} \int_{t=0}^{T} |\bar{c}_e^T(t) - c_e^T(t)| \, dt \tag{17}$$

In addition, we report a normalized mean absolute error by dividing MAE by the maximum observed experimental extravascular concentration.

*Simulation of IV-DDS.* We assumed that IV-DDS are administered before $t=0$, and are distributed uniformly within systemic plasma before triggered release within the tumor is initiated. Any clearance or extravasation of IV-DDS is assumed negligible, which is appropriate as nanoparticle extravasation takes many hours[53]. Further, we assume that IV-DDS are completely stable (i.e., no systemic leakage) during triggered release.

For comparison to experimental data with TSL, we assumed rate of triggered release varies based on temperature data measured during in vivo studies (Fig. 4a). From time-varying temperature and in vitro measured temperature-dependent TSL release times (Table S3), we calculated a time-varying release time that was applied in the computer model. We used a cubic interpolation to model the relation between release time and temperature based on the data from Table S3. We simulated both fast (fTSL) and slow release TSL (sTSL), based on their specific temperature-dependent release times. Mean absolute error was again calculated according to Eq. (17).

*Identification of key parameter ratios.* During IV-DDS based delivery, both plasma and EES drug concentration converge towards a plateau concentration (see Fig. 4c, d). Since plasma and EES concentrations are equal once this plateau is reached, the total extracted drug amount (Eq. (11)) is now zero. The EES concentration ($c_e^T$) is constant and equal to the plateau concentration ($c_e^T = c_{plateau}$), and from Eq. (11) we can then derive the following equation:

$$c_{plateau} = \int_{\xi=0}^{1} c_p^T(\xi) d\xi \tag{18}$$

Eq. (18) depends exclusively on $c_p^T$ (Eq. (9)), which further depends on Eq. (10). We also assume that the systemic concentration of IV-DDS encapsulated drug ($c_{p\_DDS}^S(t)$) is constant during triggered release, based on our assumption of a small release volume above. Since $c_e^T$ ($=c_{plateau}$) is constant as well in Eq. (9), the only model parameters that dictate the plateau concentration (Eq. (18)) considering Eq. (9) and Eq. (10) are: (1) the ratio of IV-DDS release time to tissue transit time ($t_{rel}/TT$), and (2) the ratio of vascular permeability-surface area product to plasma perfusion ($PS/F_p$). Accordingly, we defined two indices based on these two ratios: the release index $R.I.=t_{rel}/TT$, and the permeability index $P.I.= PS/F_p$.

*Parametric studies of IV-DDS.* We performed ~1200 computer simulations where the two parameter index ratios ($R.I.= t_{rel}/TT$) and ($P.I.= PS/F_p$) were varied. Except for these ratios, the model was based on parameters in Table S2. We calculated the steady-state concentration (=plateau concentration ($c_{plateau}$), see Fig. 4d) achieved for each parameter combination, and plotted a parametric map with plateau concentration indicated on a color scale (Fig. 5a). This plateau

concentration ($c_{plateau}$) represents the maximum concentration that can be delivered to the tumor interstitium (EES) for a particular parameter combination.

*Uncertainty analysis.* We performed a Monte Carlo simulation considering the uncertainty associated with all model input parameters listed in Table S4 except for derived parameters ($v_e$, $k_{av}$, $V_p^T$, $F_p$):

- $v_e$ and $k_{av}$ were not individually considered, since the product of the two is considered: $v_{e,av} = k_{av} * v_e$
- $F_p$ was not considered, since it is derived from two other parameters: $F_p = v_p^T / TT$
- $V_p^T$ was not considered, since it is derived from two other parameters: $V_p^T = VT * v_p^T$

We performed simulations both for unencapsulated drug and for IV-DDS (i.e., sTSL and fTSL). All parameters were considered distributed based on a log-normal distribution; prior studies have suggested using a log-normal distribution for biological data to address limitations of the normal distribution, such as negative parameter values[81]. Based on this Monte Carlo simulation, the standard deviation was calculated for output variables $c_p^T$ and $c_e^T$. This standard deviation was represented as error bars in Fig. 2d, e and Fig. 4d, e. We confirmed in a convergence analysis that the employed sample size ($n = 1000$) was adequate.

*Sensitivity analysis.* We performed a global sensitivity analysis considering all the parameters listed in Table S4, except for the derived parameters ($v_e$, $k_{av}$, $V_p^T$, $F_p$), as we did for the uncertainty analysis. The analysis was carried out for unencapsulated drug, as well as for fTSL as model of an IV-DDS (i.e. with $t_{rel}$ as additional parameter). For two parameters where no experimental data was available to estimate variability ($t_{rel}$, $VT$), we assumed a standard deviation equal to 10% of the mean (parameters are marked by '*' in Table S4). We performed a global sensitivity analysis via variance decomposition based on Sobol's method[82], using an approximation described earlier[83]. We used latin hypercube sampling to create randomized input parameter samples, and adapted functions from the publicly available SAFE toolbox to perform the sensitivity analyses[84]. The models were run with 390k parameter iterations for unencapsulated drug, and with 420k parameter iterations for IV-DDS (the difference is due to the latter having one additional parameter to consider, $t_{rel}$). The sensitivity analyses were performed on the Clemson Univ. Palmetto cluster supercomputer, utilizing 80 cores, 156 GB memory, and 17–20 h of computing time. As objective function we used the root-mean-square error (RMSE) of the EES concentration ($c_e^T$), using as reference ($= c_{e,REF}$) the EES concentration $c_e^T$ based on the mean for all individual input parameters, calculated over T = 20 min:

$$Err = \sqrt{\frac{1}{T} \int_{t=0}^{T} \left( c_e^T - c_{e,REF}^T \right)^2 dt} \qquad (19)$$

One limitation of the sensitivity analysis is the assumption of independent parameters. While this assumption is likely appropriate for the majority of the parameters, it is possible that there is some dependency between some of the parameters (e.g. the transit time may depend on the vascular fraction).

**Reporting summary**. Further information on research design is available in the Nature Research Reporting Summary linked to this article.

## Data availability
The datasets generated during the current study are available from the corresponding author on reasonable request.

## Code availability
Image processing was performed by custom scripts in MeVisLab v2.7.3 and Matlab 2020a. Computer models were solved in Matlab 2020a. Code used for the computational modeling studies is available from the corresponding author on reasonable request.

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

## Acknowledgements

We thank the National Institutes of Health Medical Arts team for creating Fig. 1 and Fig. S9. We thank O. Friman, M. Schwenke, and T. Preusser from the Fraunhofer MeVis Institute for Medical Image Computing for assistance with image-processing and motion compensation methods. Supported by NIH grant R01CA181664 (D.H.) and NIH C06 RR018823 from the Extramural Research Facilities Program of the National Center for Research Resources (MUSC).

## Author contributions

Conceptualization and methodology were performed by T.L.M.T. and D.H. Writing of the original draft, review, and editing were performed by T.L.M.T., M.R.D., A.L.B.S., and D.H. Investigation and formal analysis were performed by T.L.M.T., A.L.B.S., S.Z., A.L.B. S., M.A., L.L., and D.H. Funding acquisition, supervision, and project administration was performed by T.L.M.T. and D.H.

## Competing interests

The authors declare the following competing interests: M.D. is an employee of Boston Scientific, Marlborough, MA. The remaining authors declare no competing interests.
