## [Peer Review File · Communications Biology]

Reviewers' Comments:

Reviewer #2:

Remarks to the Author:

In this article, the authors describe a new computational model to describe thermally triggered drug delivery systems. A significant strength of this work is the model parameters were determined and validated experimentally based on measurements in the in vivo models used. This new computational model could provide a very interesting means to design nanomaterials for locally triggered drug delivery. It is generally well described, organized, and written. I have the following comments on the manuscript which I believe should be addressed prior to publication:

1. In the introduction, the authors point out cancer nanomedicines can reach the tumor sites by passive means through the EPR effect. However, this has been contested in recent years given this effect seems largely limited to animal models with rapidly growing tumors as compared to human tumors which may grow over several years. That being said, I think it would be to the authors benefit to point out the shortcomings of passive DDS [see for example, Sindhvani et al, Nature Materials (2020)].

2. The authors use a thermosensitive liposomal formulation engineered with slow and fast release kinetics upon exposure to heat. However, some additional information should be provided on these formulations such as the size, loading, and surface charge of these formulations. In addition, the authors note the formulations are "stable at 37°C" in the supplement but provide no additional data to show how this is defined and over what time period they are stable.

3. I would recommend in the figures, the computational and experimental results be plotted together as opposed to side-by-side to make direct comparison easier. For example in figure 2, Panels C and D could be modified to show tumor plasma in C & tumor EES in D with the experimental data shown as points and the computational data shown as a line. Similarly in figure 4, I would suggest the Panels D and E be modified to show experimental and computational results for sTSL in D and fTSL in E. This would be important to show here to indicate where the model is more or less comparable to the experimental results.

4. For Figure 4, could additional images be shown of the intravital images over time for both sTSL and fTSL? At present it is also unclear what system is being shown, some additional labeling on the figure or clarification in the legend would be helpful.

5. As it relates to the aforementioned results in Figure 4 on slow versus fast release, the authors clearly show the benefit of rapid release for intra-tumoral delivery. In addition, the peak drug concentrations and overall dynamics are reasonably well captured. However, there are some differences in the width of the peaks (i.e. duration of peak drug exposure) when comparing the experiments to the model. Generally, there is very limited discussion on how the model compares to the data for IV-DDS and it is taken to be accurate based on the free drug validation experiments (Fig. 3). In addition, the authors do not make any indications what else could be considered to improve the model's predictive capabilities.

6. In Figure 5, the authors investigate key parameters that they believe dictate IV-DDS showing how delivery can be optimized based on tissue permeability and release rate. However, this is a very challenging figure to understand based on the number of demarcations, symbols, color scales, etc. included. I would recommend these key ratios be given their own symbols as indices of interest. For example, PS/Fp and trel/TT could be defined as 'tissue permeability' and 'release rate' indices so that the reader can more easily see what is represented on the x- and y-axis.

Reviewer #3:

Remarks to the Author:

Intravascular triggered drug delivery systems (IV-DDS) for local drug delivery include various stimuli responsive nanoparticles that release the associated agent in response to internal (e.g., pH and enzymes) or external stimuli (e.g., temperature, light, ultrasound, electromagnetic fields, x-

rays). The authors developed a computational model to simulate IV-DDS drug delivery, for which they quantified all model parameters in vivo using a fluorescent dye. The validation of the model was via quantitative intravital microscopy studies with temperature-sensitive liposomes as example IV-DDS. The IV-DDS delivery kinetics was interpreted based on this model with the parameters of IV-DDS of drug and target tissue for optimal delivery being identified. The results reported are interesting and worthy to be considered for publication. However, the organization and presentation of the paper may have made it confusing to the readers and there is no mentioning of ways to implement the model for other types of stimuli such as pH, which limits the potential for translational applications. There are also some minor grammatical revisions needed. Therefore, the reviewer would suggest the following changes before the paper can be accepted and believe that when better presented the paper will influence thinking in the field and be of interest to the community and the wider field.

Detailed comments:

1. The paper is presented as majorly focused on computer simulations, however, there are only two equations mentioned about in the main manuscript and there is a lack of reference to many details of the model, which were only found in the supplementary information after the reviewer got confused and dig into it. In order for the model to make sense and be of potential use by other researchers, it is probably necessary that some of the details of the model be shown and explained in the main manuscript.
2. In order for the model to have greater impacts, it is suggested that the authors consider mentioning of the other ways to trigger release, e.g., by pH (Guo, et al., *Adv. Mater* 30: 1705436, 2018; Guo, et al., *Progress in Mater. Sci.*, 107: 100599, 2020) and light (Zhu, et al., *JACS* 140, 11: 4062, 2018; Barhoumi, et al., *Nano Lett.*, 14: 3697, 2014), as well as novel ways to monitor the release of actual drugs, e.g., through an emissive hydrazone photochrome (*Chemical Science*, 11: 3016, 2020), and then elaborate on how to potentially employ this model for those applications. These will add significantly to the potential impacts of this work.
3. References should be cited for "The case of complete extraction ($EF \sim 1$, equivalent to $F_p \ll PS$) is commonly termed "perfusion-limited transport", while the case of incomplete extraction ($EF \ll 1$, equivalent to $PS \ll F_p$) is termed "permeability limited transport". " and "initiated by the release 6 trigger present in the target region (e.g. tumor)."
4. Consideration of cellular uptake will be critical for applications in in vivo experiments, so comments on how to potentially modify the model accordingly would be appreciated.
5. There is not Summary or Conclusions, although this could be due to the format requirement of the journal—worth checking.
6. Minor grammatical changes needed:
 - 1) On Page 2, "but encapsulated drug is released inside the vessels and distributes readily into the targeted tissue" should be "but the encapsulated drug is released inside the vessels and readily distributed into the targeted tissue";
 - 2) "x-ray" should be "X-Ray".
 - 3) On Page 3, "up to 25 higher tumor drug uptake" is not correct.
 - 4) On Page 3, there shouldn't be two "often" in "...often limited by often unknown parameter".
 - 5) On Page 5, "in case of high permeability" should likely be "in the case of high permeability".
 - 6) On Page 7, "and considering further in vitro measured..." should likely be "and considered further in vitro measured..."
 7. "e.g." should be changed to "e.g.,", for example, in the abstract.

RESPONSE TO INITIAL REVIEW

We would like to thank both reviewers for their careful review and for their helpful comments for improving the manuscript. Below we individually address each of the comments and describe the changes we made in response. In our resubmission, a manuscript version is included where all changes are highlighted.

Editorial Comments

Please address **all** concerns raised by reviewers. In particular, we request that further technical details are needed where requested (reviewer 2, point 2 and reviewer 3, point 1) and that more intravital images are provided that show both sTSL and fTSL over time (reviewer 2, point 4). We also strongly encourage an expansion of the discussion (reviewer 2, point 5). We encourage some discussion about how this model can be altered to reflect changes in cellular uptake (reviewer 3, point 4).

Response

We added the additional details on the liposomal formulation (reviewer 2, point 2 and reviewer 3, point 1); we added additional intravital images for both sTSL and fTSL over time (reviewer 2, point 4) and also now provide the whole movies for both sTSL and fTSL time series as supplementary material. We added a discussion on the differences between computer model and experiment (reviewer 2, point 5), and discussion on how to implement cell uptake in the model (reviewer 3, point 4). Below we respond in more detail to each of the reviewer's comments and concerns.

REVIEWER #2

General Comments

In this article, the authors describe a new computational model to describe thermally triggered drug delivery systems. A significant strength of this work is the model parameters were determined and validated experimentally based on measurements in the in vivo models used. This new computational model could provide a very interesting means to design nanomaterials for locally triggered drug delivery. It is generally well described, organized, and written. I have the following comments on the manuscript which I believe should be addressed prior to publication:

Detailed Comments

Comment #1

In the introduction, the authors point out cancer nanomedicines can reach the tumor sites by passive means through the EPR effect. However, this has been contested in recent years given this effect seems largely limited to animal models with rapidly growing tumors as compared to human tumors which may grow over several years. That being said, I think it would be to the authors benefit to point out the shortcomings of passive DDS [see for example, Sindhvani et al, Nature Materials (2020)].

Response

We added brief descriptions of the limitations of EPR-based delivery in the introduction and discussion, citing the reference above in addition to another relevant reference. The text added in the introduction is:

“While passive accumulation may be enhanced via targeting moieties and better retention, there is increasing consensus that alternate approaches are necessary as the EPR effect appears less effective in human tumors, and is highly heterogenous within and between tumors¹⁸⁻²⁰.”

In the discussion, we added the following text:

“Most drug delivery systems (DDS) are based on passive (EPR-based) or affinity targeting for accumulation^{8,14,16}, but there is a need for alternative approaches due the inherent limitations of EPR-based delivery^{6,18,20}.”

Comment #2

The authors use a thermosensitive liposomal formulation engineered with slow and fast release kinetics upon exposure to heat. However, some additional information should be provided on these formulations such as the size, loading, and surface charge of these formulations. In addition, the authors note the formulations are “stable at 37°C” in the supplement but provide no additional data to show how this is defined and over what time period they are stable.

Response

We added additional data in the supplementary materials section for both liposomal formulations for: TSL size, polydispersity index, zeta potential, loading efficiency, and stability at 37 °C (Fig. S10B).

Comment #3

I would recommend in the figures, the computational and experimental results be plotted together as opposed to side-by-side to make direct comparison easier. For example in figure 2, Panels C and D could be modified to show tumor plasma in C & tumor EES in D with the experimental data shown as points and the computational data shown as a line. Similarly in figure 4, I would suggest the Panels D and E be modified to show experimental and computational results for sTSL in D and fTSL in E. This would be important to show here to indicate where the model is more or less comparable to the experimental results.

Response

We added a Figure in both Figure 2 and Figure 4 directly comparing EES concentration between computer model and intravital data, for both sTSL and fTSL (new Figs. 2E and 5E). We do believe that it is also informative to directly see a comparison between plasma and EES concentration, since this visualizes the transport kinetics between plasma and EES, so we kept this comparison as we had in the initial submission. However, the new figures show where the model is more or less comparable to the experimental results in a direct comparison.

Comment #4

For Figure 4, could additional images be shown of the intravital images over time for both sTSL and fTSL? At present it is also unclear what system is being shown, some additional labeling on the figure or clarification in the legend would be helpful.

Response

We added additional figures for both sTSL and fTSL in Figure 4, and label the time at which the images were obtained in relation to the other graphs. As mentioned earlier, we also now include movies of the whole time series for a fTSL and a sTSL dataset as additional supplementary material.

Comment #5

As it relates to the aforementioned results in Figure 4 on slow versus fast release, the authors clearly show the benefit of rapid release for intra-tumoral delivery. In addition, the peak drug concentrations and overall dynamics are reasonable well captured. However, there are some differences in the width of the peaks (i.e. duration of peak drug exposure) when comparing the experiments to the model. Generally, there is very limited discussion on how the model compares to the data for IV-DDS and it is taken to be accurate based on the free drug validation experiments (Fig. 3). In addition, the authors do not make any indications what else could be considered to improve the model’s predictive capabilities.

Response

We added a more detailed description of the differences between model results and the *in vivo* study, and include possible reasons for this deviation that indicate how the model may be amended to address these differences. Below is the text we added in response to the manuscript:

“Compared to the intravital data, the computer model predicted a lower peak with wider plateau for fTSL. Further, for both sTSL and fTSL the computer model predicted a more rapid washout after heating stops that was less pronounced *in vivo*, especially for sTSL (Fig. 5E). Possible explanations for these observed differences include: (1) presence of an additional mechanism relevant at low concentrations (e.g., systemic leakage from TSL, cell uptake, or non-linear binding to plasma and/or tissue constituents); (2) TSL temperature dependence of release kinetics may differ *in vivo* from the *in vitro* data on which the model is based; (3) inaccuracies in tumor temperature measurements; (4) changes in tumor parameters in response to hyperthermia (e.g. perfusion).”

Comment #6

In Figure 5, the authors investigate key parameters that they believe dictate IV-DDS showing how delivery can be optimized based on tissue permeability and release rate. However, this is a very challenging figure to understand based on the number of demarcations, symbols, color scales, etc. included. I would recommend these key ratios be given their own symbols as indices of interest. For example, PS/Fp and trel/TT could be defined as ‘tissue permeability’ and ‘release rate’ indices so that the reader can more easily see what is represented on the x- and y-axis.

Response

We added the two indices as suggested by this reviewer in Fig 5A and throughout the manuscript text. In addition, we added a new Fig. 5B, where we indicate relevant regions in the parametric map of Fig. 5A to help the reader interpret Fig. 5A.

REVIEWER #3

General Comments

Intravascular triggered drug delivery systems (IV-DDS) for local drug delivery include various stimuli responsive nanoparticles that release the associated agent in response to internal (e.g., pH and enzymes) or external stimuli (e.g., temperature, light, ultrasound, electromagnetic fields, x-rays). The authors developed a computational model to simulate IV-DDS drug delivery, for which they quantified all model parameters *in vivo* using a fluorescent dye. The validation of the model was via quantitative intravital microscopy studies with temperature-sensitive liposomes as example IV-DDS. The IV-DDS delivery kinetics was interpreted based on this model with the parameters of IV-DDS of drug and target tissue for optimal delivery being identified. The results reported are interesting and worthy to be considered for publication. However, the organization and presentation of the paper may have made it confusing to the readers and there is no mentioning of ways to implement the model for other types of stimuli such as pH, which limits the potential for translational applications. There are also some minor grammatical revisions needed. Therefore, the reviewer would suggest the following changes before the paper can be accepted and believe that when better presented the paper will influence thinking in the field and be of interest to the community and the wider field.

Detailed Comments

Comment #1

The paper is presented as majorly focused on computer simulations, however, there are only two equations mentioned about in the main manuscript and there is a lack of reference to many details of the model, which were only found in the supplementary information after the reviewer got confused and dig into it. In order for the model to make sense and be of potential use by other researchers, it is probably necessary that some of the details of the model be shown and explained in the main manuscript.

Response

We moved the relevant model details, equations, model parameter tables and model assumptions from the supplementary materials to the methods section in the main manuscript as suggested by this reviewer, including additional descriptions of the model and definitions of the parameters.

Comment #2

In order for the model to have greater impacts, it is suggested that the authors consider mentioning of the other ways to trigger release, e.g., by pH (Guo, et al., *Adv. Mater* 30: 1705436, 2018; Guo, et al., *Progress in Mater. Sci.*, 107: 100599, 2020) and light (Zhu, et al., *JACS* 140, 11: 4062, 2018; Barhoumi, et al., *Nano Lett.*, 14: 3697, 2014), as well as novel ways to monitor the release of actual drugs, e.g., through an emissive hydrazone photochrome (*Chemical Science*, 11: 3016, 2020), and then elaborate on how to potentially employ this model for those applications. These will add significantly to the potential impacts of this work.

Response

We added the suggested references in the introduction and discussion as additional ways to trigger release, and monitoring of drug release and drug uptake. In addition, we included a new section in the discussion describing how the computer model could be adapted for other triggered delivery systems:

“The presented computational model considers the release trigger (e.g., temperature time course, Fig. 4A) and how IV-DDS release kinetics varies with trigger magnitude (e.g., temperature dependence of TSL release, Fig. S10-S12). Adaptation of the present computer model to other types of IV-DDS would thus require the knowledge of the release trigger dynamics at the target site (e.g., ultrasound intensity for microbubbles, or light fluence rate for light-sensitive IV-DDS), and data on how IV-DDS release kinetics varies with magnitude of the trigger. Since these data only represent input variables to the model, in most cases no changes of the underlying model equations would be required. If there is interaction of the IV-DDS with the microvasculature, the IV-DDS transit time may differ from plasma transit time and would need to be determined independently. In case the 0th order release kinetics is not adequate for a particular IV-DDS, other release kinetics can be substituted in the model by modifying the release term (see Suppl. Mat).”

Comment #3

References should be cited for “The case of complete extraction ($EF \sim 1$, equivalent to $F_p \ll PS$) is commonly termed “perfusion-limited transport”, while the case of incomplete extraction ($EF \ll 1$, equivalent to $PS \ll F_p$) is termed “permeability limited transport”. ”and “initiated by the release trigger present in the target region (e.g. tumor).”

Response

We added appropriate references for the two statements indicated by the reviewer.

Comment #4

Consideration of cellular uptake will be critical for applications in in vivo experiments, so comments on how to potentially modify the model accordingly would be appreciated.

Response

We added a section in the discussion describing how cellular uptake may be integrated based on available mathematical models describing cell uptake. Below is the text that we added:

“To include cellular drug uptake in the computer model, an equation would need to be added that describes cell uptake based on amount of free drug available in the EES. The kinetics of cellular uptake depends on the drug and may also vary by cell type. For common agents such as doxorubicin, platinum drugs (cisplatin, oxaliplatin, carboplatin) and paclitaxel, mathematical models describing cell uptake that were derived from *in vitro* studies are available^{70,71}. Such cell uptake models may be readily integrated in this computational model, similar to prior studies^{40,45,56}.

Comment #5

There is not Summary or Conclusions, although this could be due to the format requirement of the journal—worth checking.

Response

We checked and the journal guidelines do not include any summary or conclusions. However, the final paragraph in the discussion includes a brief summary with conclusions.

Comment #6

Minor grammatical changes needed:

- 1) On Page 2, “but encapsulated drug is released inside the vessels and distributes readily into the targeted tissue” should be “but the encapsulated drug is released inside the vessels and readily distributed into the targeted tissue”;
- 2) “x-ray” should be “X-Ray”.
- 3) On Page 3, “up to 25 higher tumor drug uptake” is not correct.
- 4) On Page 3, there shouldn’t be two “often” in “...often limited by often unknown parameter”.
- 5) On Page 5, “in case of high permeability” should likely be “in the case of high permeability”.
- 6) On Page 7, “and considering further in vitro measured...” should likely be “and considered further in vitro measured...”
7. “e.g.” should be changed to “e.g.,”, for example, in the abstract.

Response

We thank the reviewer for the suggested corrections and corrected all manuscript sections as indicated by the reviewer.

Reviewers' Comments:

Reviewer #2:

Remarks to the Author:

The authors have addressed my previous comments and I have nothing further to added.

Reviewer #3:

Remarks to the Author:

The revised manuscript addresses the previous concerns, so it is recommended for publication after minor revisions. The reviewer suggests that the authors go through the language carefully and make corrections/improvements as deemed necessary. For example, "Unfortunately" should be "Unfortunately,", permeability index (P.I.) was defined multiple times, and " due the inherent limitations" should be "due to the inherent limitations".

RESPONSE TO REVISION REVIEW

We would like to thank both reviewers for their careful review. Below we address the remaining minor issues indicated by the second reviewer (Reviewer #3).

REVIEWER #2

The authors have addressed my previous comments and I have nothing further to added.

REVIEWER #3

The revised manuscript addresses the previous concerns, so it is recommended for publication after minor revisions. The reviewer suggests that the authors go through the language carefully and make corrections/improvements as deemed necessary. For example, "Unfortunately" should be "Unfortunately,", permeability index (P.I.) was defined multiple times, and " due the inherent limitations" should be "due to the inherent limitations".

Response

We corrected the minor issues noted above and removed superfluous definitions of the permeability index (P.I.).